# Exploratory analysis of the accuracy of age-based maximal heart rate equations across cardiorespiratory fitness levels

Joel Martin[1,2]*, Bryndan Lindsey[3], Courtney Gerrity[1], Jatin Ambegaonkar[1]

**1** Sports Medicine Assessment Research & Testing (SMART) Laboratory, George Mason University, Fairfax, Virginia, United States of America, **2** Center for the Advancement of Well-Being, George Mason University, Fairfax, Virginia, United States of America, **3** State Key Laboratory of Mechanical System and Vibration, School of Mechanical Engineering, Shanghai Jiao Tong University, Shanghai, China

* jmarti38@gmu.edu

## Abstract

### Introduction

Maximal heart rate (MHR) is a key measure for cardiorespiratory exercise prescription yet is often estimated using age-based prediction equations. The accuracy of these equations may vary by individual characteristics, including cardiorespiratory fitness (CRF), but limited research has examined predictive accuracy across CRF levels. Therefore, we evaluated the accuracy of seven commonly used MHR prediction equations in adults with varying CRF to assess whether prediction error differs by fitness level.

### Materials and methods

Data from 230 healthy adults (76% male, mean age 38.5 ± 12.3 years) who completed maximal graded exercise tests between 2019 and 2024 were analyzed retrospectively. Predicted MHR values were calculated using the Fox, Tanaka, Gellish, Arena, Åstrand, Nes, and Fairbairn equations. Linear mixed-effects models (LMM) tested the influence of $VO_2max$ and its interaction with prediction equation on error, with sex included as a covariate. Estimated marginal means and slopes were extracted, with pairwise contrasts adjusted by the Tukey method. Prediction equation accuracy was evaluated by comparing predicted and measured MHR using Bland-Altman analyses, and metrics including mean absolute error (MAE), root mean square error (RMSE), and intraclass correlation coefficients (ICC).

### Results

LMM indicated a significant main effect of prediction equation on error ($p < 0.001$) and a significant equation × $VO_2max$ interaction ($p = 0.015$), though neither sex ($p = 0.49$) nor $VO_2max$ ($p = 0.18$) alone influenced error. The conditional $R^2$ for the LME model

**Data availability statement:** All relevant data are within the manuscript and its Supporting information files. The dataset generated and analyzed during the current study has been deposited in the Open Science Framework and is publicly available under DOI https://doi.org/10.17605/OSF.IO/VDG92 (version 1.0). The dataset can be accessed at: https://osf.io/vdg92/?view_only=47d195f2b4d94641a1b54d-d2a67cca2c.

**Funding:** The author(s) received no specific funding for this work.

**Competing interests:** The authors have declared that no competing interests exist.

was 0.70, with a marginal $R^2$ of 0.02. Post-hoc linear regressions showed higher $VO_2$max was associated with greater prediction error for several equations in males, but not females, with a small amount of variance explained ($R^2 \leq 0.06$). Agreement analyses indicated small mean biases across equations (–3 to +6 bpm) but wide limits of agreement (~±18–24 bpm). Arena, Tanaka and Gellish equations showed the lowest MAE and RMSE. Among the equations, Fox showed the most stable performance across MHR ranges, being the only formula without proportional bias across the sample.

## Discussion

The findings indicate that CRF had only a limited influence on MHR prediction error, with small associations observed in males but not females, reinforcing age as the primary determinant of MHR. Although some equations (e.g., Tanaka, Gellish, Arena, Fox) performed better than others across agreement metrics, none demonstrated high individual level accuracy, which highlights a lack of precision when estimating MHR for exercise prescription and monitoring purposes. Future work should explore more individualized modeling approaches, though adjusting for CRF alone may not substantially improve prediction accuracy in healthy adults.

## Introduction

Maximal heart rate (MHR) in beats per minute (bpm) is a fundamental parameter in exercise physiology, commonly used to assess cardiorespiratory fitness (CRF) [1] and prescribe training intensity [2]. Direct measurement of MHR requires a graded exercise test (GXT) taken to volitional exhaustion, which may pose logistical challenges and potential health risks, particularly for individuals with underlying medical conditions or limited exercise tolerance [3]. As a result, age-based MHR prediction equations are widely used as a safer, more convenient alternative to directly measuring MHR [1]. The most common prediction equation is the simple 'MHR = 220 – age' formula, developed by Fox and colleagues [4]; although it has been criticized for high inter-individual variability [3,5]. In response, numerous alternative equations have been proposed to improve prediction accuracy due to high inter-individual variability in MHR across different populations [1,6–8].

Several widely used MHR prediction equations are derived from large population studies or specific subgroups [1,6–8]. Notable examples include the Tanaka equation (MHR = 208 − 0.7 × age) introduced by Tanaka et al. [1] based on a meta-analysis of 18,712 subjects, the Gellish equation (MHR = 207 − 0.7 × age) from a longitudinal fitness study in 2007 with participants (n = 908) of a broad age and fitness spectrum [6], and the Nes equation (MHR = 211 − 0.64 × age) derived from a large (n = 3320) Norwegian cohort [9]. Other widely used formulas include Åstrand's formula (216.6 − 0.84 × age) [10], and Arena's formula (209.3 − 0.72 × age) which were proposed in the context of cardiac rehabilitation and functional capacity testing to gauge effort [11].

While these MHR prediction equations rely primarily on age, cardiorespiratory responses to exercise may differ between sexes due to physiological and hormonal differences [12]. Accordingly, some MHR models have attempted to account for sex differences [7]. For instance, Fairbairn's equations are sex-specific for males and females (e.g., male: MHR = 208 − 0.8 × age; female: MHR = 201 − 0.63 × age) with the inclusion of sex intended to improve accuracy for female individuals [7]. With numerous MHR prediction formulas available and derived from different samples [1,6–8], it is evident that no single equation perfectly fits all as each carries assumptions based on the characteristics of its derivation sample [13]. Importantly, across all formula, age has emerged as the predominant factor to predict MHR.

One major limitation of all age-based MHR prediction formulas is that age explains only part of the variability in MHR [1,13,14]. While MHR declines about 0.5 to 1 beats per year in adulthood [14], other factors such as sex [7], genetics [15], and CRF level [16] may also influence MHR. Notably, there has been debate about the effect of CRF and/or endurance training on MHR [16]. Higher CRF achieved through endurance training is generally associated with enhanced stroke volume and autonomic regulation, increased vagal tone at rest [17], and significantly lower resting heart rates [18]. Some evidence indicates that chronically aerobic-trained people show a slight reduction in MHR compared to age-matched untrained peers [16], potentially due to cardiac remodeling and increased parasympathetic influence that accompanies aerobic conditioning [16]. However, others have observed increases in MHR after detraining or inconsistent changes with training [19,20]. Zavorsky [16] and Carter et al. [21] have detailed these conflicting findings, noting instances where MHR in previously sedentary adults decreased with exercise training, only to increase again upon training cessation [16,21]. Interestingly, Lach et al. [22] found that adding CRF and body composition to prediction models only marginally improved accuracy ($R^2$ = 0.22 vs. 0.19 for age alone), suggesting limited influence of CRF on MHR for physically active adults. To our knowledge this finding has yet to be explored in subsequent studies. Thus, while age remains the most informative single covariate in population models of MHR, the influence of CRF level on MHR and on the accuracy of prediction is not well resolved.

In summary, the literature indicates that age-based MHR predictions are convenient but lack accuracy on the individual level [13] and it remains uncertain whether any equation is more suitable based on CRF level. The potential error introduced by CRF level represents an important gap in the literature, as knowledge of consistent over or underestimation by a specific equation would affect the ability for exercise professionals to make more informed decisions when selecting the most appropriate formula. To address this gap, the present study undertook an exploratory analysis of the accuracy of several commonly used age-predicted MHR equations across adults with varying CRF levels. In line with recent recommendations for distinguishing exploratory from confirmatory research [23,24], no *a priori* hypotheses were tested. The study findings may provide preliminary insights of value to exercise science practitioners, clinicians, and researchers who frequently utilize MHR prediction equations in individuals with varying levels of CRF when devising cardiorespiratory training and exercise programs for their clients and patients.

## Materials and methods

### 2.1. Participants and design

This study was a retrospective analysis of de-identified graded exercise tests (GXT) data collected between 2019 and 2024 at a university exercise physiology laboratory. The data were accessed on January 23, 2025 for research purposes. A total of 230 adults (174 males, 56 females) who had performed GXTs during this period were included. Participant ages ranged from 18 to 68 years (mean ± SD: 38.5 ± 12.3 years) with a range of CRF levels. The participants were individuals from the local community who voluntarily participated in the laboratory's human performance GXT testing services. In most cases, they were recreational athletes of varying levels seeking estimates of $VO_2$max and MHR to guide training. Others underwent testing for general health or fitness assessment informational purposes. Inclusion criteria for the analysis were: age ≥ 18; completion of a GXT to volitional fatigue with valid attainment of $VO_2$max (see below); and no acute medical issues at the time of testing. The study protocol was reviewed and approved by the George Mason University

Institutional Review Board (IRB #: 1665548) and participants provided informed consent prior to testing. Because this was a retrospective analysis of an existing dataset, no *a priori* sample size calculation was conducted. As a retrospective exploratory study, we analyzed all eligible records without an a priori power analysis, following recommendations to justify sample size by using the full accessible dataset and to transparently acknowledge the absence of prospective calculations [25]. The exploratory nature of this study is intended to generate preliminary insights and identify patterns that can inform future confirmatory studies that can plan sample sizes for desired accuracy or a smallest effect size of interest [24].

## 2.2. Graded exercise test protocol

All participants performed a maximal GXT on a motorized treadmill (Desmo, Woodway, Waukesha, WI, USA) to determine $VO_2$max and MHR. Tests were conducted by trained technicians following standard laboratory procedures. Participants were advised to refrain from heavy exercise and stimulants on the day of the test and provided basic health screening information before starting. The protocol consisted of a 3-minute warm-up followed by a continuous incremental test. During warm-up, participants began with a brisk walk (~3.0 mph, 0–1 min), transitioned to a light jog (~4–5 mph, 1–2 min), and progressed to their self-selected "working pace" (2–3 min). The working pace was typically ~5–6 mph but was adjusted individually to reflect a pace the participant self-reported could maintain for 15–20 minutes. At minute 3, the first stage began at 0% grade. Thereafter, treadmill grade increased by 2% every 2 minutes (e.g., 2% at min 5, 4% at min 7, 6% at min 9, etc.). Once a 10% grade was reached, further increases were made by small increments in speed (~0.2–0.3 mph) rather than grade, to preserve running form. The test followed GXT self-paced recommendations [26] and was designed to elicit volitional exhaustion within ~8–12 minutes, excluding the warm-up. Throughout the test, heart rate (HR) was measured using a telemetry HR monitor (H10, Polar, Polar Electro, Kempele, Finland), which is shown to have superior signal quality relative to other comparable devices [27]. Expired gases were collected using a calibrated metabolic cart (Parvo Medics TrueOne 2400; Parvo Medics, Sandy, UT, USA) to determine oxygen uptake. The metabolic system was calibrated prior to each test according to manufacturer guidelines using standardized gas concentrations and volume flow calibration. Each test continued until the participant reached volitional exhaustion (e.g., gait instability, participant request to stop) or met termination criteria. Achievement of $VO_2$max (termination criteria) was verified by meeting at least two of the following criteria: (1) respiratory exchange ratio (RER) ≥ 1.10 at peak exercise; (3) heart rate plateauing near age-predicted MHR (within ~10 bpm) or reaching ≥90% of age-predicted MHR; and/or (3) rating of perceived exertion (RPE) ≥ 19 on a 6–20 Borg scale [3]. All participants included in the analysis satisfied these criteria, indicating equivalent effort across the sample [3]. MHR measured was defined as the highest heart rate value attained during the GXT.

## 2.3. Maximum heart rate prediction equations

The performance of seven common age-based MHR prediction equations were evaluated. The equations chosen for their frequent use in both research and practice [3,13]. The equations were:

- Fox [4]: MHR = 220 − age
- Tanaka [1]: MHR = 208 – 0.7 × age
- Gellish [6]: MHR = 207 – 0.7 × age
- Arena [11]: MHR = 209.3 – 0.72 × age
- Åstrand [10]: MHR = 216.6 − 0.84 × age
- Nes [9]: MHR = 211 − 0.64 × age
- Fairbairn [7]: MHR = 208 − 0.8 × age for male; MHR = 201 − 0.63 × age for female

## 2.4. Statistics

The data was exported from the metabolic cart computer software into Excel (Microsoft Corporation, Redmond, WA, USA) Nfor further analysis. All statistical analyses and visualizations were performed using R (version 4.2.1, R Core Team, Vienna, Austria). Statistical significance was set to $p < 0.05$ for all analyses with adjustments as appropriate as described below. The dataset supporting this study is openly available on the Open Science Framework at https://doi.org/10.17605/OSF.IO/VDG92. All variables required for analysis (age, sex, MHR, and VO$_2$max) were complete with no missing values. Therefore, no participants were excluded due to incomplete demographic or physiological data.

Outlier detection was performed on prediction error (Prediction Error = MHR$_{predicted}$ − MHR$_{measured}$) and absolute error values. Outliers were flagged as values exceeding ±3 standard deviations from the mean [28]. This pre-specified threshold was used as a quality-control screen to limit inclusion of likely artifacts from HR telemetry, such as signal dropouts or chest strap slippage [27]. Raw data from flagged cases were reviewed and 3 participants' data were removed for MHR values flagged as outliers resulting in a final analytical sample of N = 230. Normality of the primary continuous variables (age, VO$_2$max, MHR, and RER) was assessed using the Shapiro–Wilk test and inspection of Q–Q plots. To further examine the distributional properties, kernel density plots were generated for the overall sample and stratified by sex.

Descriptive statistics were calculated for the full sample and stratified by sex. Sex differences in continuous variables were assessed using non-parametric Mann–Whitney U tests due to violations of normality assumptions. Effect sizes, with 95% confidence intervals (CI), for group differences were calculated as Cohen's $d$, with $d = 0.2, 0.5, 0.8$ indicating small, medium, and large effects, respectively [29]. To assess the relationships between MHR versus age we regressed measured MHR on the equation-predicted MHR (simple linear model: MHR$_{measured}$ ~ MHR$_{predicted}$) and reported coefficients of determination ($R^2$) to summarize the proportion of variance explained. To examine whether the relationship between age and relative VO$_2$max differed by sex, scatterplots stratified by sex were generated with locally estimated scatterplot smoothing (LOESS) curves. Linear regression models including sex, age, and the sex × age interaction term were fit to test for sex-specific differences in slopes. $R^2$ were computed for each sex to quantify the strength of association between age and VO$_2$max. Slope contrasts were performed to compare the estimated age-related decline in VO$_2$max between males and females.

To evaluate the effects of CRF on MHR prediction equations, we quantified error using absolute differences between predicted and measured values. Linear mixed-effects models (LMMs) were fit using the *lme4* package in R. Fixed effects included Equation (seven levels: Fox, Tanaka, Gellish, Arena, Åstrand, Nes, Fairbairn), VO$_2$max (continuous), their interactions, and sex (as a covariate). VO$_2$max values were mean centered prior to analysis so that model intercepts reflected prediction accuracy at the sample average, and to reduce multicollinearity between main effects and interaction terms. Subjects were modeled with random intercepts to account for repeated measures across equations. Estimated marginal means and simple slopes were extracted from the models using the *emmeans* package, with pairwise post hoc contrasts adjusted using the Tukey method to control for multiple comparisons. LMMs were selected over repeated-measures ANOVA because they better accommodate unbalanced data, relax sphericity assumptions, and allow simultaneous inclusion of continuous covariates and interaction terms, providing a more flexible framework for modeling interindividual variability [30].

To further evaluate whether CRF modified the accuracy of MHR prediction equations, we fit separate ordinary least squares regression models with absolute error as the dependent variable and VO$_2$max as the predictor. The regression slope quantified whether individuals with higher CRF exhibited systematically greater or lower prediction errors. Regression models were fit in R using the lm() function. For each model, the slope estimate with 95% CI, $p$-value, $R^2$, and residual standard error (RMSE) were calculated.

Bland–Altman analyses were conducted for each prediction equation against measured MHR to evaluate systematic and proportional bias, stratified by sex. Systematic bias was quantified as the mean difference (MHR$_{predicted}$ − MHR$_{measured}$), and 95% limits of agreement (LOA) were calculated as the mean difference ± 1.96 SD of the differences. Proportional bias

was assessed by linear regression of the prediction error on measured MHR, with significance indicating a non-constant bias across the range of values. To further evaluate the agreement between MHR$_{predicted}$ and MHR$_{measured}$, several metrics were computed for each prediction equation. These included mean absolute error (MAE), and root-mean-square error (RMSE) to quantify systematic and absolute differences. Agreement between MHR$_{predicted}$ and MHR$_{measured}$ was further evaluated using an intraclass correlation coefficient (ICC, two-way mixed-effects model, ICC [1,3]). Each metric was calculated for the entire sample and separately for males and females. ICC values were interpreted using the following thresholds: $< 0.50 = poor$, $0.50–0.74 = moderate$, $0.75–0.89 = good$, and $≥0.90 = excellent$ reliability [31]. To facilitate comparison, equations were ranked from 1 (best) to 7 (worst) for each metric within each group.

## Results

### 3.1. Participant characteristics

Table 1 summarizes the characteristics of the 230 participants. Shapiro–Wilk tests indicated that VO$_2$max ($W = 0.993$, $p = 0.31$) was normally distributed, whereas Age ($W = 0.969$, $p < 0.001$), MHR ($W = 0.985$, $p = 0.017$), and respiratory exchange ratio ($W = 0.915$, $p < 0.001$) were not normally distributed. The data distributions are visualized in Fig 1. The sample was 75.7% male. Sex differences ($p < 0.05$) were observed for height, body mass, and relative VO$_2$max, with males presenting significantly greater values and effect sizes ranging from medium to large. No significant sex differences were detected for age, MHR, or respiratory exchange ratio (all $p > 0.05$). Descriptive statistics for the overall sample and by sex are presented in Table 1.

### 3.2. Relationships of maximum heart rate and VO$_2$max with age

Simple linear regressions of measured versus predicted MHR showed comparable variance explained across all equations (Fig 2), with $R^2$ values ranging from 0.40 to 0.45. Despite differences in slope and intercept across the age-based formulas, the non–sex-specific equations (Fox, Tanaka, Åstrand, Gellish, Nes, and Arena) all explained a similar

**Table 1. Overall participant ($n = 230$) characteristics.**

| | Total (n = 230) | Males (n = 174) | Females (n = 56) | p-value | Effect Size Cohen's d |
|---|---|---|---|---|---|
| Age (yrs) | MN (SD): 38.5 (12.3) MD (IQR): 37 (18.75) Range: [18, 68] | MN (SD): 38.6 (12.1) MD (IQR): 36.5 (18.8) Range: [18, 68] | MN (SD): 38.1 (12.9) MD (IQR): 38.0 (21.8) Range: [18, 62] | 0.72 | 0.04 [-0.26, 1.86] |
| Height (m) | MN (SD): 175.1 (8.3) MD (IQR): 175.0 (9.3) Range: [149.9,198.1] | MN (SD): 177.7 (6.9) MD (IQR): 176.5 (9.1) Range: [152, 198.1] | MN (SD): 167.0 (7.3) MD (IQR): 168.3 (9.8) Range: [149.9, 180.3] | < 0.001 | 1.53 [1.20, 1.86] |
| Mass (kg) | MN (SD): 79.7 (15.3) MD (IQR): 79.4 (18.5) Range: [48.5, 141.9] | MN (SD): 84.1 (13.8) MD (IQR): 84.1 (13.8) Range: [53.7, 141.9] | MN (SD): 65.9 (11.1) MD (IQR): 65.9 (11.1) Range: [48.5, 109.9] | < 0.001 | 1.38 [1.05, 1.71] |
| VO$_2$max (mL/kg-min) | MN (SD): 45.0 (9.3) MD (IQR): 45.3 (12.2) Range: [20.7, 65.3] | MN (SD): 46.4 (9.1) MD (IQR): 46.6 (12.8) Range: [21.2, 65.3] | MN (SD): 39.9 (12.5) MD (IQR): 39.9 (12.5) Range: [20.7, 61.3] | < 0.001 | 0.66 [0.35, 0.97] |
| RER | MN (SD): 1.16 (0.05) MD (IQR): 1.2 (0.1) Range: [1.10, 1.39] | MN (SD): 1.16 (0.05) MD (IQR): 1.15 (0.07) Range: [1.10, 1.39] | MN (SD): 1.16 (0.05) MD (IQR): 1.15 (0.07) Range: [1.10, 1.29] | 0.33 | 0.10 [-0.14, 0.47] |
| MHR | MN (SD): 180.4 (12.0) MD (IQR): 181.0 (16.0) Range: [143, 209] | MN (SD): 180.1 (11.5) MD (IQR): 181 (15) Range: [144, 209] | MN (SD): 181.3 (13.2) MD (IQR): 184 (17) Range: [143, 199] | 0.33 | 0.16 [-0.40, 0.20] |

Note: Values in Total, Males, and Females columns are mean (MN) and standard deviation (SD), with Medians (MD), Interquartile Ranges (IQR) and minimum and maximum values (in brackets). Values in Effect Size column are Cohen's d with 95% confidence intervals in brackets.

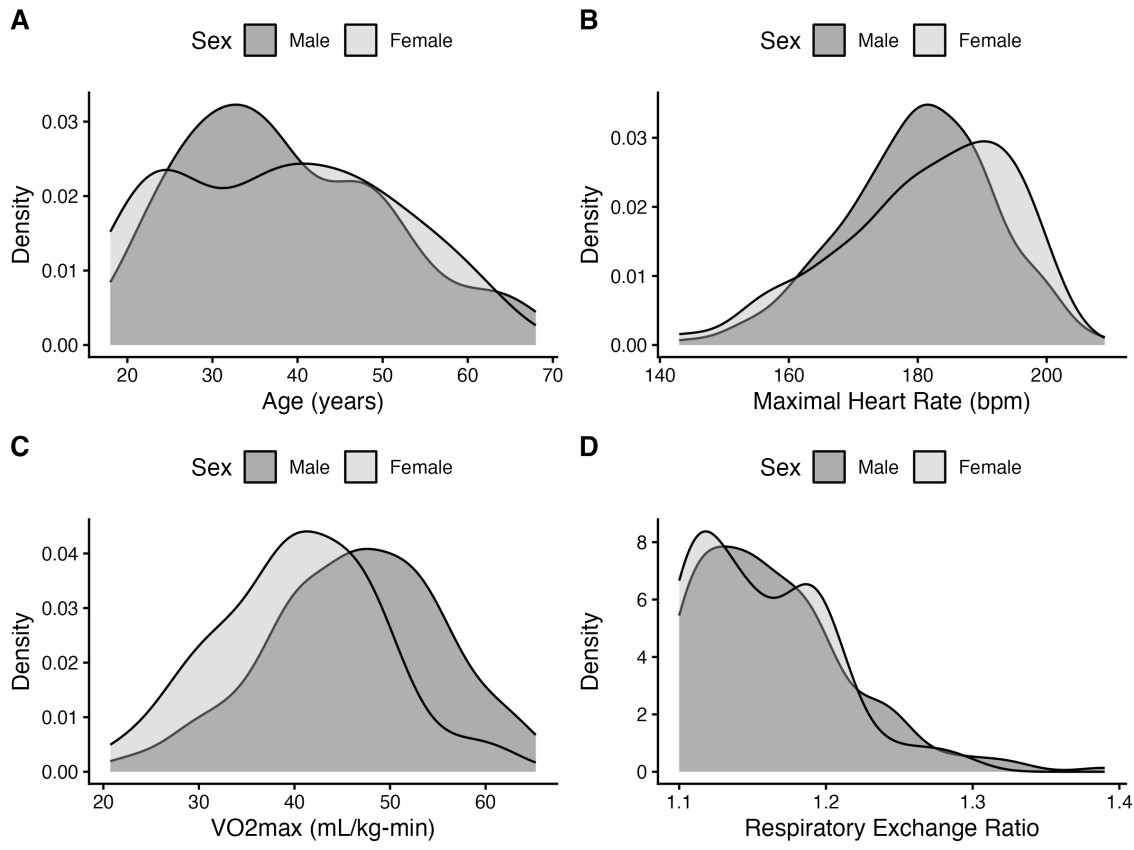

**Fig 1. Kernel density plots of age, maximum heart rate, VO2max, and respiratory exchange ratio.**

proportion of variance in measured MHR ($R^2 = 0.42$). This similarity reflects the fact that each equation relies primarily on age as the predictor, producing highly collinear estimates.

Linear regression models (Fig 3) revealed that age was negatively associated with relative VO$_2$max in both males and females. Among males, age explained 24% of the variance in VO$_2$max ($R^2 = 0.236$, $p < 0.001$), with an average decline of −0.37 mL/kg-min per year (95% CI [−0.47, −0.27]). Among females, age explained 18% of the variance ($R^2 = 0.18$, $p = 0.001$), with an average decline of −0.28 mL/kg-min per year (95% CI [−0.45, −0.12]). The Age × Sex interaction was not statistically significant ($\beta = 0.085$, 95% CI [−0.11, 0.28], $p = 0.386$), indicating that the rate of decline in VO$_2$max with age did not differ significantly between males and females.

### 3.3. Influence of VO$_2$max on predicted maximum heart rate

LMM results indicated a significant main effect of prediction equation on error, $F(6, 1389) = 5.83$, $p < 0.001$, and a significant equation × VO$_2$max interaction, $F(6, 1389) = 2.63$, $p = 0.015$ (Table 2). Neither sex ($p = 0.49$) nor VO$_2$max ($p = 0.18$) had an independent effect on predicted error. Post hoc contrasts revealed that, relative to the Fox equation (reference), the Tanaka ($\beta = −1.03$, 95% CI [−1.63, −0.43], $p < 0.001$), Gellish ($\beta = −0.98$, 95% CI [−1.58, −0.38], $p = 0.001$), and Arena ($\beta = −0.98$, 95% CI [−1.58, −0.38], $p = 0.001$) equations were associated with significantly lower absolute error. No other between-equation differences were significant. Only one significant slope difference was detected in the equation × VO$_2$-max interaction, with the Fairbairn equation demonstrated a stronger negative association between VO$_2$max and error ($\beta = −0.079$, $p = 0.016$). Given the significant equation x VO$_2$max interaction, we investigated simple slopes of VO$_2$max

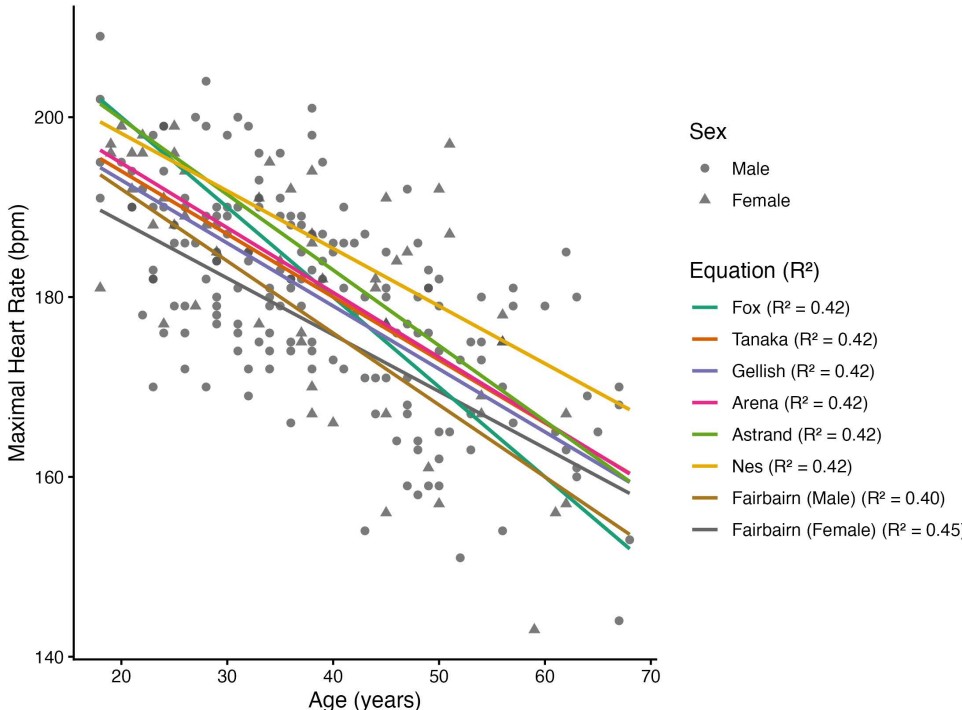

**Fig 2. Measured maximum heart rate versus age with age-based prediction equations.** Note: Measured maximal heart rate as a function of age, with predicted values from seven age-based equations superimposed. Colored lines represent the prediction equations, with associated $R^2$ values denoting model fit to the measured data.

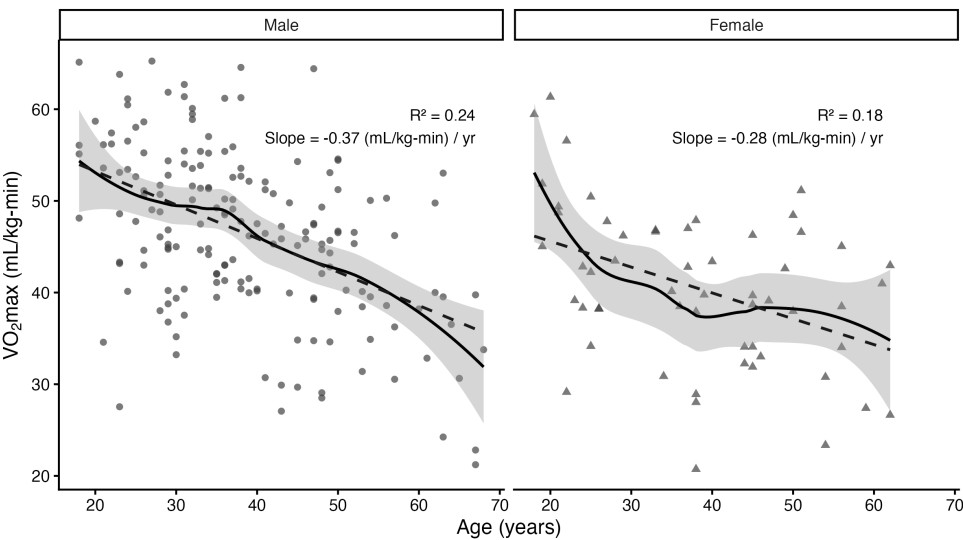

**Fig 3. Age-related decline in relative VO₂max stratified by sex.** Note: Scatterplots with LOESS smoothing (solid black line) and linear regression fits (dashed line) are shown for males and females. The shaded area represents the 95% confidence interval around the LOESS curve.

**Table 2. Linear Mixed-Effects Model Examining the Effects of Prediction Equation, VO$_2$max, and Sex on Absolute Error of Maximum Heart Rate Prediction.**

| Predictor Equation | Estimate ($\beta$) | 95% CI [Lower, Upper] | SE | df | t | p-value |
|---|---|---|---|---|---|---|
| Intercept (ref: Fox) | 8.33 | [7.43, 9.23] | 0.46 | 313 | 18.2 | <0.001 |
| Tanaka | −1.03 | [-1.63, -0.43] | 0.31 | 1389 | −3.37 | <0.001 |
| Gellish | −0.98 | [-1.58, -0.38] | 0.31 | 1389 | −3.21 | 0.001 |
| Arena | −0.98 | [-1.58, -0.38] | 0.31 | 1389 | −3.20 | 0.001 |
| Åstrand | −0.17 | [-0.77, 0.43] | 0.31 | 1389 | −0.54 | 0.590 |
| Nes | 0.19 | [-0.41, 0.79] | 0.31 | 1389 | 0.62 | 0.530 |
| Fairbairn | −0.25 | [-0.85, 0.35] | 0.31 | 1389 | −0.81 | 0.420 |
| Sex (ref: Female) | 0.57 | [-1.06, 2.20] | 0.83 | 211 | 0.69 | 0.490 |
| VO$_2$max | 0.06 | [-0.02, 0.14] | 0.04 | 616 | 1.50 | 0.130 |
| Tanaka × VO$_2$max | −0.02 | [-0.09, 0.04] | 0.03 | 1389 | −0.66 | 0.510 |
| Gellish × VO$_2$max | −0.03 | [-0.10, 0.03] | 0.03 | 1389 | −0.97 | 0.330 |
| Arena × VO$_2$max | −0.01 | [-0.08, 0.05] | 0.03 | 1389 | −0.42 | 0.680 |
| Åstrand × VO$_2$max | 0.05 | [-0.02, 0.11] | 0.03 | 1389 | 1.39 | 0.170 |
| Nes × VO$_2$max | −0.00 | [-0.07, 0.06] | 0.03 | 1389 | −0.09 | 0.930 |
| Fairbairn × VO$_2$max | −0.08 | [-0.14, -0.01] | 0.03 | 1389 | −2.42 | 0.016 |

**Model statistics:** Conditional $R^2$ = 0.697, Marginal $R^2$ = 0.015., CI = Confidence Interval, SE = Standard Error.

**ANOVA results:** Equation ($p$ < 0.001), Equation × VO$_2$max ($p$ = 0.015), Sex ($p$ = 0.49), VO$_2$max main effect ($p$ = 0.18).

within each equation. Among these exploratory comparisons, only the Åstrand equation exhibited a significant positive slope ($\beta$ = 0.106, 95% CI [0.027, 0.185], $p$ = 0.009), indicating that error increased as VO$_2$max increased. However, the conditional $R^2$ for the model was 0.70, with a marginal $R^2$ of 0.02, indicating that variance was dominated by subject-level random effects rather than fixed effects.

Regression analyses stratified by sex revealed that VO$_2$max was significantly associated with absolute error in several prediction equations among males but not females. In males, higher VO$_2$max was linked to greater prediction error for the Fox ($\beta$ = 0.10 bpm per mL/kg-min, $p$ = 0.033), Tanaka ($\beta$ = 0.10, $p$ = 0.03), Arena ($\beta$ = 0.10, $p$ = 0.02), Åstrand ($\beta$ = 0.16, $p$ = 0.001), and Nes ($\beta$ = 0.11, $p$ = 0.05) equations. The Gellish equation showed a positive slope that approached significance ($\beta$ = 0.08, $p$ = 0.055). In contrast, the Fairbairn equation did not show a significant relationship ($\beta$ = 0.03, $p$ = 0.57). Among females, slopes were small and non-significant across all equations ($\beta$ range: –0.05 to 0.03, all $p$ > 0.61). The proportion of variance explained by VO$_2$max was low ($R^2$ ≤ 0.06), indicating that although CRF influenced prediction error in males, the effect size was small. Figs 4–10 present the regression slopes with 95% CIs, highlighting significant positive associations for several equations in males, but no associations in females. Notably, the 95%CI of slopes indicate overlap between sexes and thus indicate no sex differences in slope per equation.

### 3.4. Prediction equation agreement with maximum heart rate

Bland–Altman plots and corresponding statistics for the seven prediction equations are presented in Figs 11–17. Across the sample, mean biases were generally small, ranging from a slight underestimation with the Fairbairn equation (−3.26 bpm) to overestimation with the Nes equation (+6.02 bpm). LOA were wide across all equations (approximately ±18–24 bpm), indicating substantial individual-level variability. Evidence of proportional bias was present for most equations, with stronger effects observed for Tanaka, Gellish, Arena, Nes, and Fairbairn (all $p$ < 0.01). In these cases, predicted MHR increasingly underestimated measured values at higher heart rates. In contrast, the Fox equation demonstrated neither significant mean nor proportional bias, suggesting relatively stable performance across the range of measured MHR.

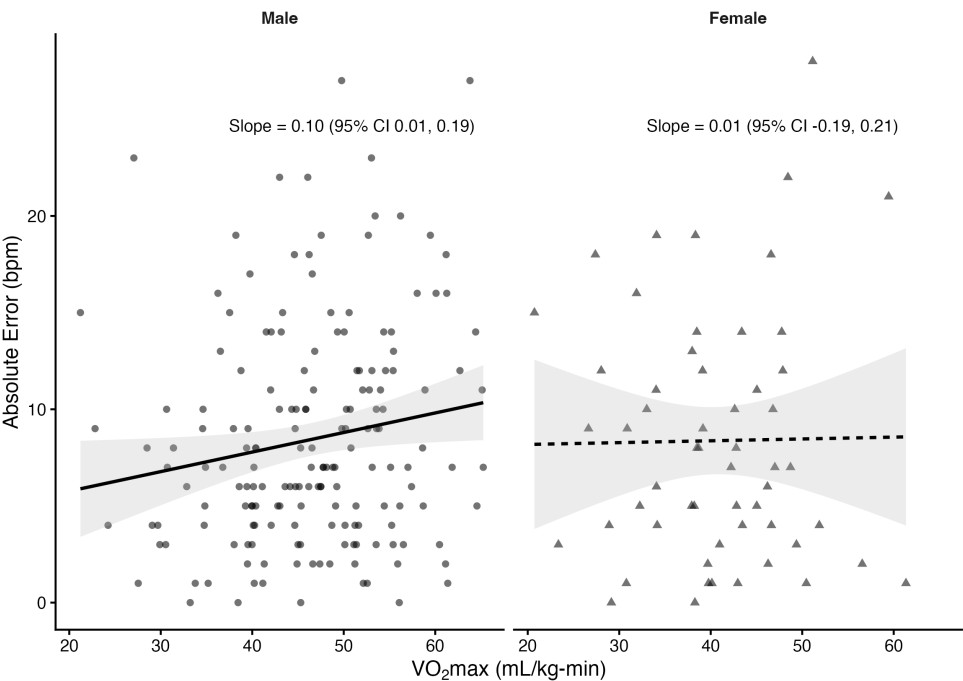

**Fig 4. Relationship between VO₂max and maximum heart rate prediction error for the fox equation.** Note: Points represent individual observations, and solid lines represent sex-specific ordinary least squares regression fits with shaded 95% confidence intervals. Regression slopes with 95% CIs are displayed at the right side of each panel.

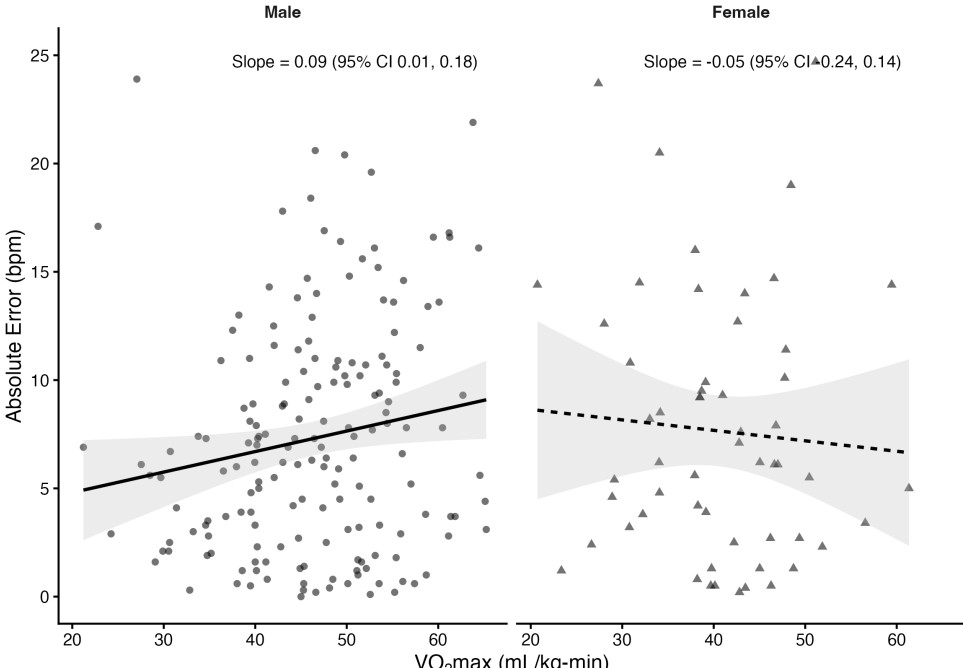

**Fig 5. Relationship between VO₂max and maximum heart rate prediction error for the Tanaka equation.** Note: Points represent individual observations, and solid lines represent sex-specific ordinary least squares regression fits with shaded 95% confidence intervals. Regression slopes with 95% CIs are displayed at the right side of each panel.

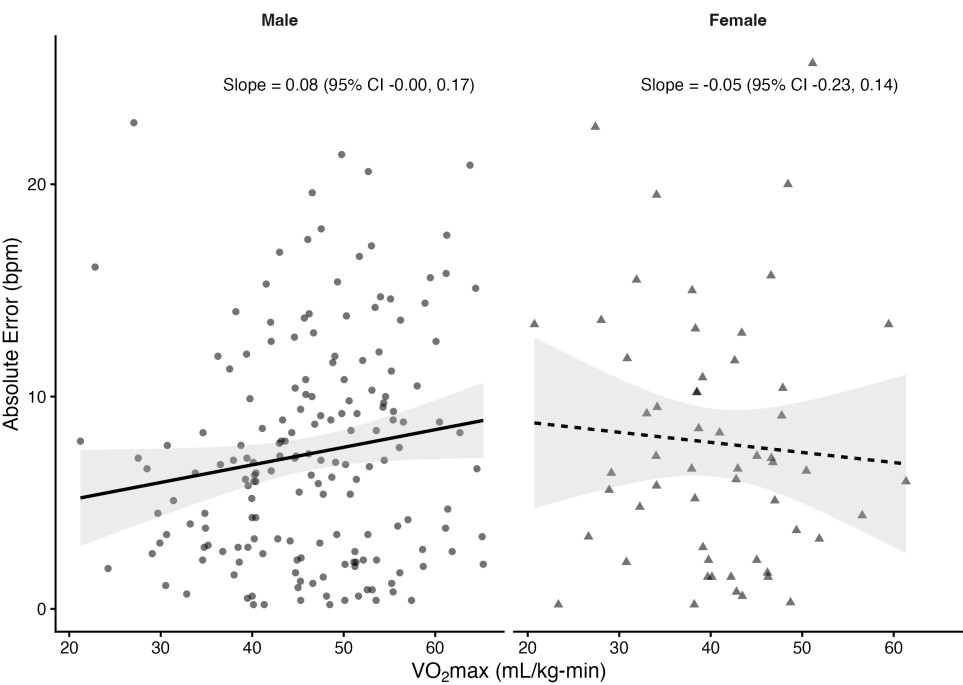

**Fig 6. Relationship between VO$_2$max and maximum heart rate prediction error for the Gellish equation.** Note: Points represent individual observations, and solid lines represent sex-specific ordinary least squares regression fits with shaded 95% confidence intervals. Regression slopes with 95% CIs are displayed at the right side of each panel.

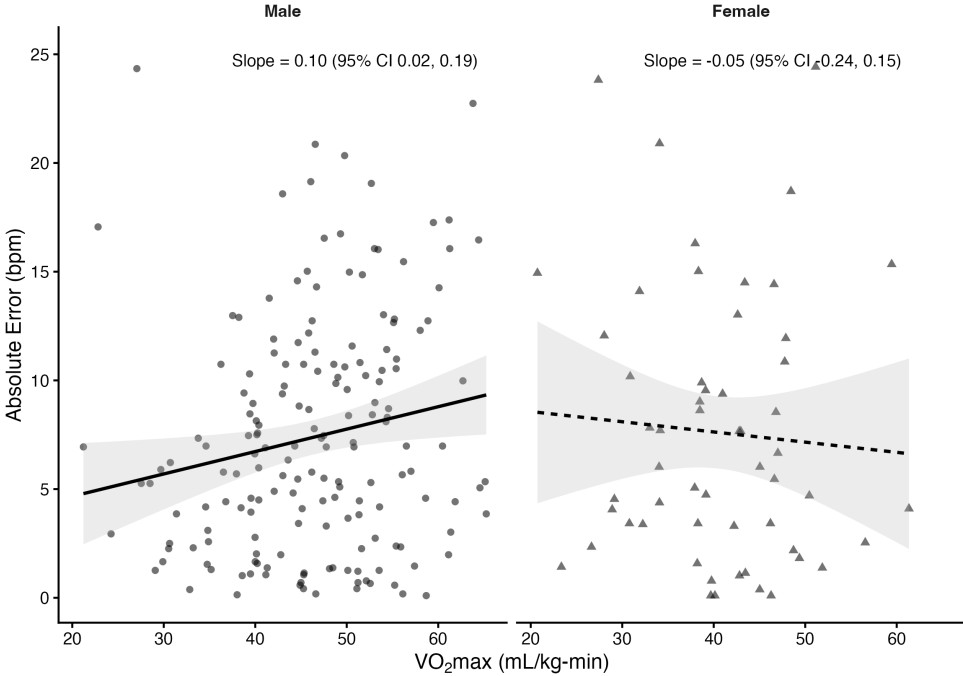

**Fig 7. Relationship between VO$_2$max and maximum heart rate prediction error for the Arena equation.** Note: Points represent individual observations, and solid lines represent sex-specific ordinary least squares regression fits with shaded 95% confidence intervals. Regression slopes with 95% CIs are displayed at the right side of each panel.

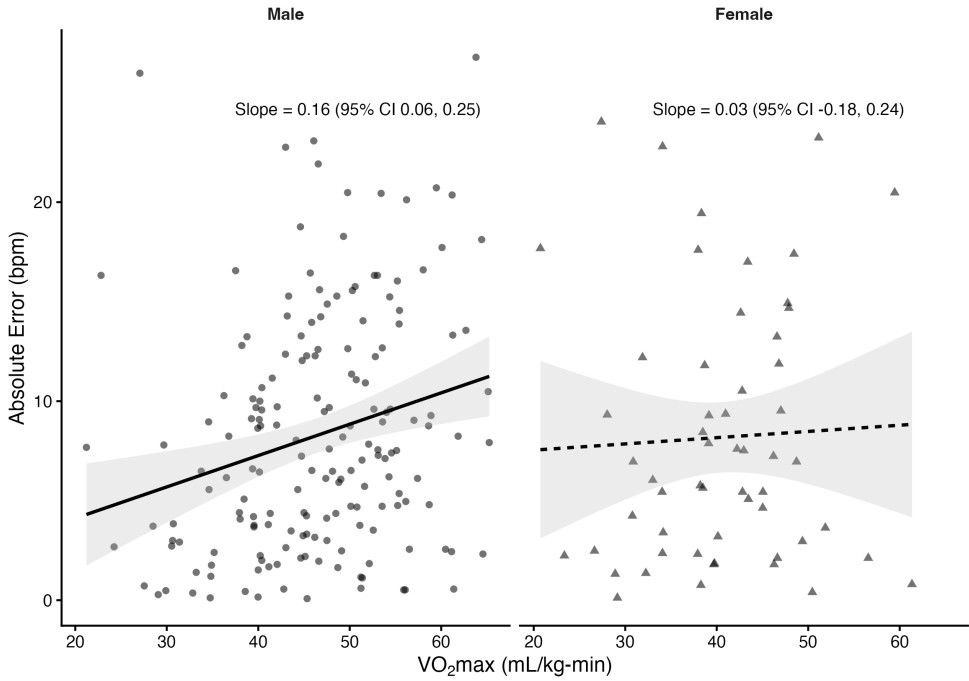

**Fig 8. Relationship between VO$_2$max and maximum heart rate prediction error for the Astrand equation.** Note: Points represent individual observations, and solid lines represent sex-specific ordinary least squares regression fits with shaded 95% confidence intervals. Regression slopes with 95% CIs are displayed at the right side of each panel.

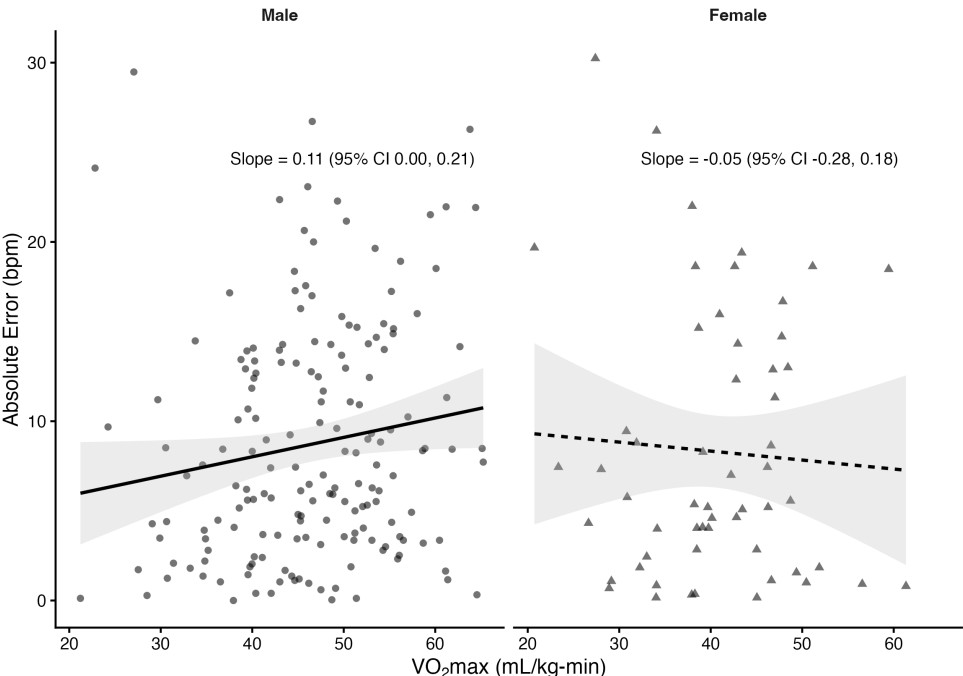

**Fig 9. Relationship between VO$_2$max and maximum heart rate prediction error for the Nes equation.** Note: Points represent individual observations, and solid lines represent sex-specific ordinary least squares regression fits with shaded 95% confidence intervals. Regression slopes with 95% CIs are displayed at the right side of each panel.

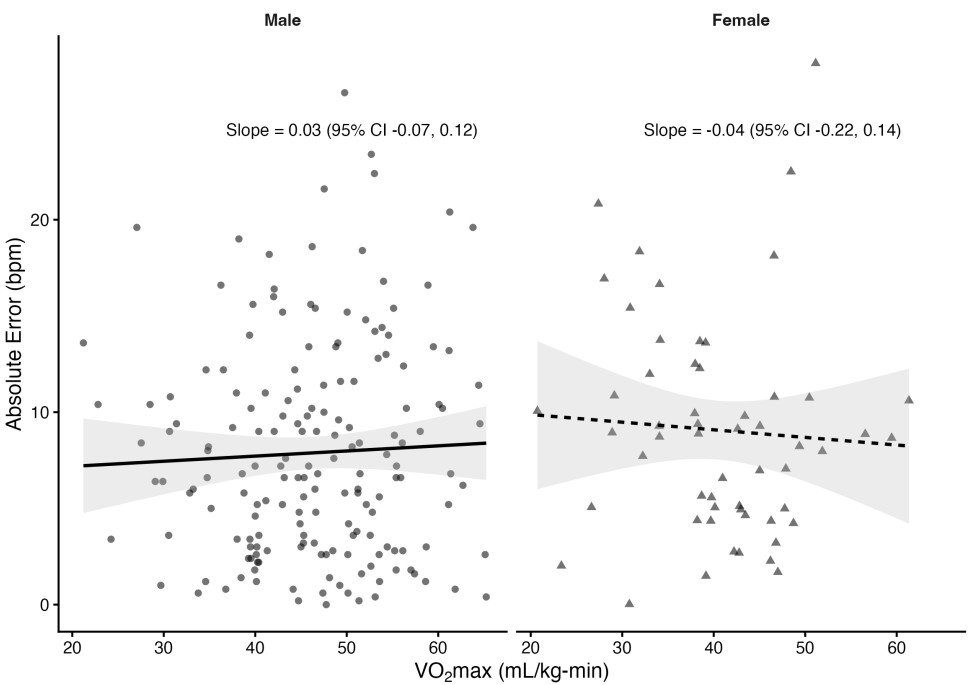

**Fig 10. Relationship between VO$_2$max and maximum heart rate prediction error for the Fairbairn equation.** Note: Points represent individual observations, and solid lines represent sex-specific ordinary least squares regression fits with shaded 95% confidence intervals. Regression slopes with 95% CIs are displayed at the right side of each panel.

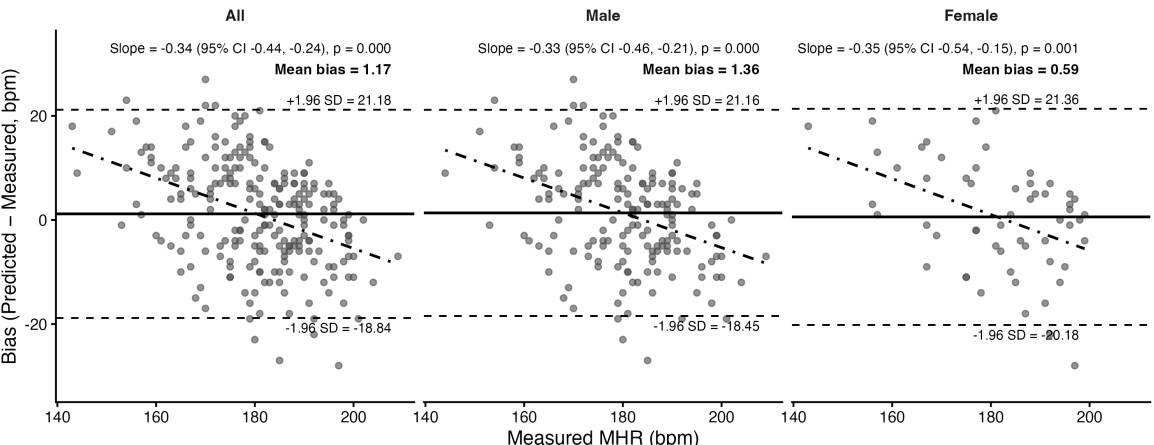

**Fig 11. Bland Altman analysis plots of maximum heart rate prediction for the Fox Equation.** Abbreviations: MHR = Maximum Heart Rate; bpm, beats per minute. The x-axis represents measured MHR in bpm, and the y-axis shows the difference between predicted and measured MHR (bias). The solid horizontal line represents the mean bias, while the upper and lower dashed lines indicate the 95% limits of agreement (LOA). The sloped dashed line shows the proportional bias estimated from linear regression, with the regression equation and p-value reported at the top of each plot.

Sex-stratified analyses revealed broadly similar patterns for males and females, though females tended to exhibit slightly wider limits of agreement and, in some cases, stronger proportional bias (e.g., Fairbairn, Nes).

To complement the Bland–Altman results, pairwise contrasts from the LMM identified several significant between-equation differences for the overall sample. Relative to Fox, the Tanaka (estimate = 1.03, 95% CI [0.43, 1.63], $p = 0.014$),

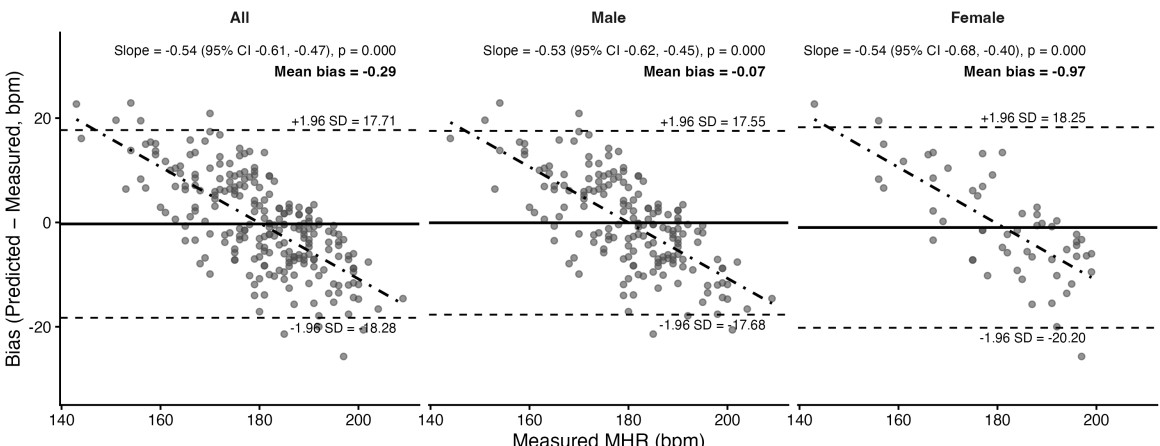

**Fig 12. Bland Altman analysis plots of maximum heart rate prediction for the Gellish equation.** Abbreviations: MHR = Maximum Heart Rate; bpm, beats per minute. The x-axis represents measured MHR in bpm, and the y-axis shows the difference between predicted and measured MHR (bias). The solid horizontal line represents the mean bias, while the upper and lower dashed lines indicate the 95% limits of agreement (LOA). The sloped dashed line shows the proportional bias estimated from linear regression, with the regression equation and p-value reported at the top of each plot.

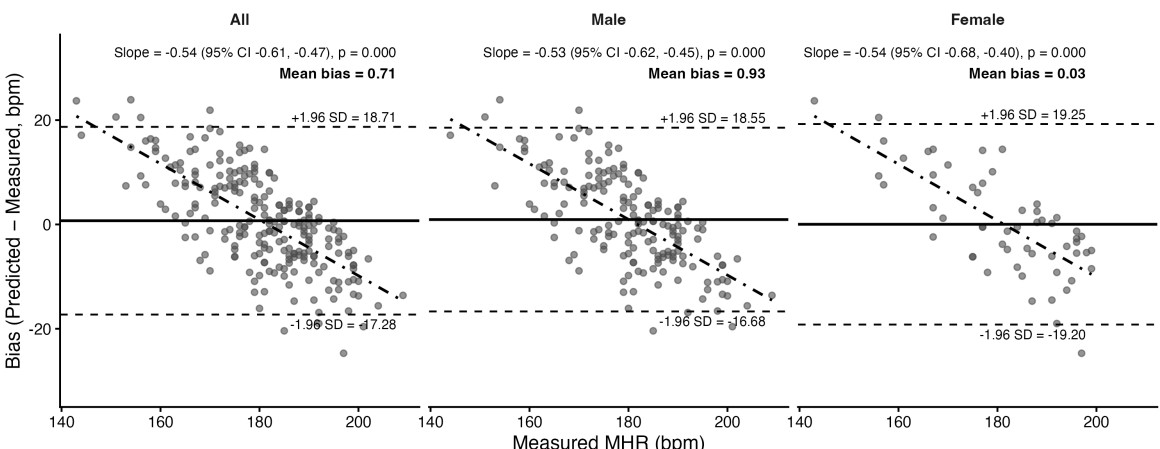

**Fig 13. Bland Altman analysis plots of maximum heart rate prediction for the Tanaka equation.** Abbreviations: MHR = Maximum Heart Rate; bpm, beats per minute. The x-axis represents measured MHR in bpm, and the y-axis shows the difference between predicted and measured MHR (bias). The solid horizontal line represents the mean bias, while the upper and lower dashed lines indicate the 95% limits of agreement (LOA). The sloped dashed line shows the proportional bias estimated from linear regression, with the regression equation and p-value reported at the top of each plot.

Gellish (0.98 [0.38, 1.58], $p = 0.023$), and Arena (0.98 [0.38, 1.58], $p = 0.024$) equations all showed significantly lower absolute error. Additionally, the Nes equation showed significantly greater absolute error from Tanaka (−1.22 [−1.83, −0.62], $p = 0.001$), Gellish (−1.17 [−1.77, −0.57], $p = 0.002$), and Arena (−1.17 [−1.77, −0.56], $p = 0.003$). No other contrasts reached significance. These results indicate that although several equations (Tanaka, Gellish, Arena) performed differently from Fox, the Nes equation was consistently distinct from the others, aligning with its tendency toward larger positive bias observed in the Bland–Altman analysis.

Agreement metrics for each prediction equation are presented in Table 3 and the relative rankings across metrics are illustrated in Figs 18–20. For the entire sample, the lowest MAE and RMSE were observed for the Tanaka (MAE: 7.40 bpm, RMSE: 9.21 bpm) and Gellish equations (MAE: 7.45 bpm, RMSE: 9.19 bpm), while the Nes equation consistently

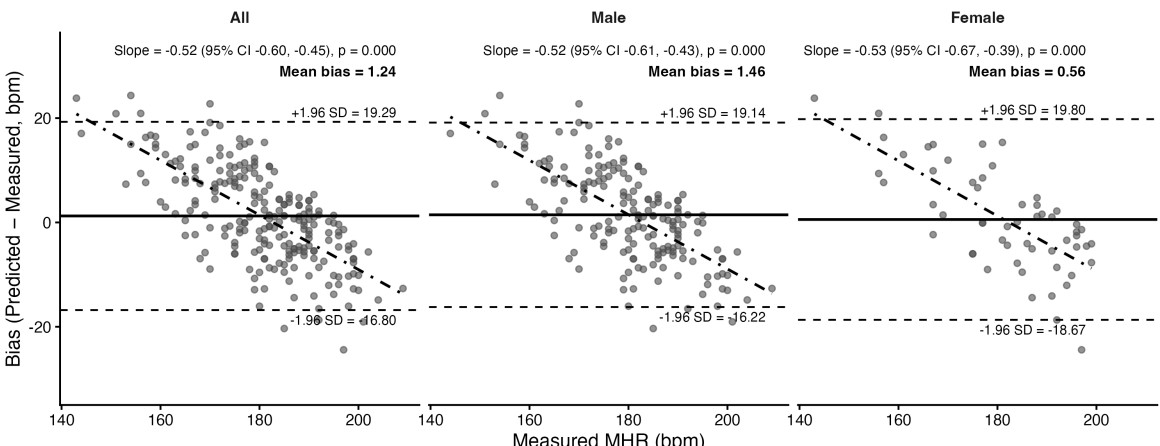

**Fig 14. Bland Altman analysis plots of maximum heart rate prediction for the Arena equation.** Abbreviations: MHR = Maximum Heart Rate; bpm, beats per minute. The x-axis represents measured MHR in bpm, and the y-axis shows the difference between predicted and measured MHR (bias). The solid horizontal line represents the mean bias, while the upper and lower dashed lines indicate the 95% limits of agreement (LOA). The sloped dashed line shows the proportional bias estimated from linear regression, with the regression equation and p-value reported at the top of each plot.

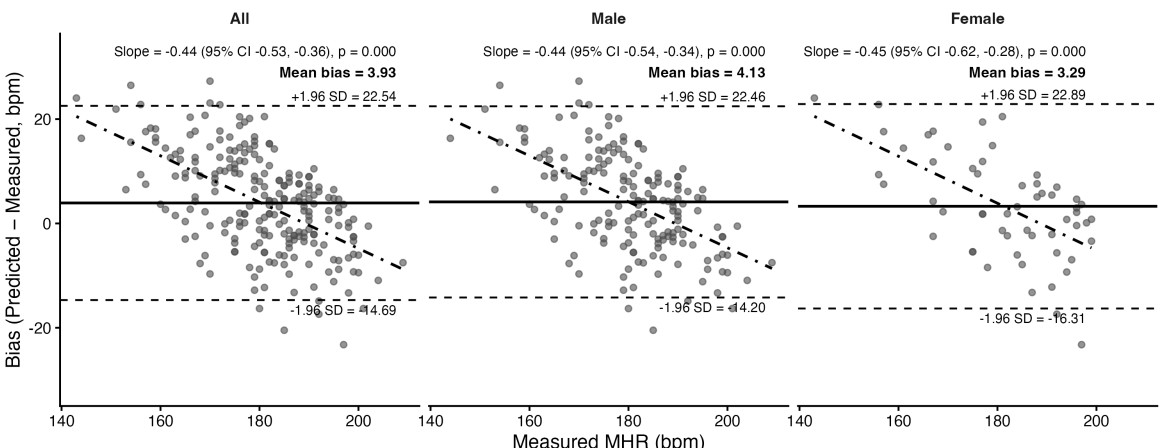

**Fig 15. Bland Altman analysis plots of maximum heart rate prediction for the Åstrand equation.** Abbreviations: MHR = Maximum Heart Rate; bpm, beats per minute. The x-axis represents measured MHR in bpm, and the y-axis shows the difference between predicted and measured MHR (bias). The solid horizontal line represents the mean bias, while the upper and lower dashed lines indicate the 95% limits of agreement (LOA). The sloped dashed line shows the proportional bias estimated from linear regression, with the regression equation and p-value reported at the top of each plot.

showed the poorest accuracy (MAE: 8.62 bpm, RMSE: 10.94 bpm). ICC(3,1) values were in the moderate (range: 0.50–0.64), with Fox demonstrating the highest values of the MHR prediction equations.

Among males, findings were similar. Tanaka (MAE: 7.32 bpm, RMSE: 9.04 bpm) and Gellish (MAE: 7.33 bpm, RMSE: 8.99 bpm) yielded the most accurate predictions, while Nes again produced the largest positive bias (+6.24 bpm). ICC was highest for Fox (0.63), but differences across equations were small (0.49–0.63).

Among females, mean bias values ranged from an underestimation of –4.30 bpm (Fairbairn) to an overestimation of +5.31 bpm (Nes). The Arena (MAE: 7.61 bpm, RMSE: 9.74 bpm) and Tanaka (MAE: 7.66 bpm, RMSE: 9.72 bpm) equations performed best on error metrics. Fox provided the ICC with measured MHR (0.67), while Nes again showed the weakest agreement (0.54).

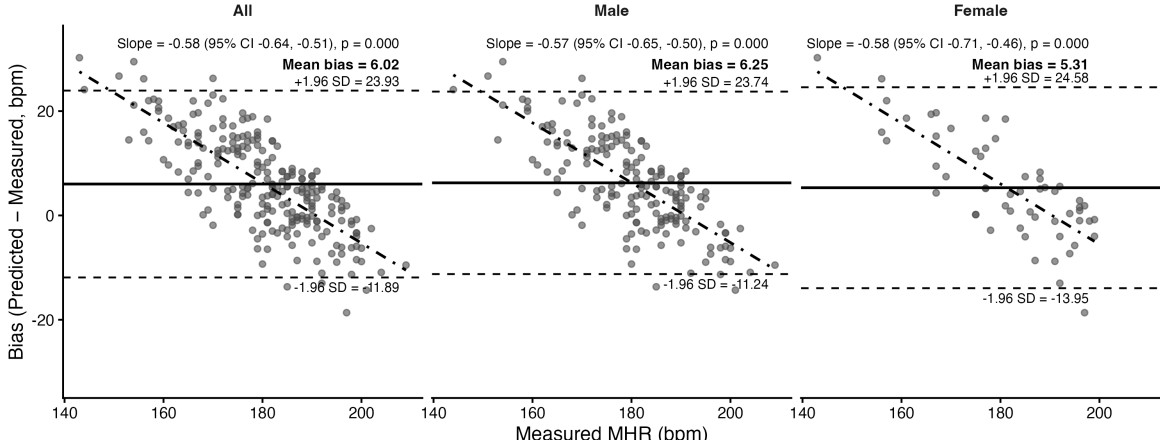

**Fig 16. Bland Altman analysis plots of maximum heart rate prediction for the Nes equation.** Abbreviations: MHR = Maximum Heart Rate; bpm, beats per minute. The x-axis represents measured MHR in bpm, and the y-axis shows the difference between predicted and measured MHR (bias). The solid horizontal line represents the mean bias, while the upper and lower dashed lines indicate the 95% limits of agreement (LOA). The sloped dashed line shows the proportional bias estimated from linear regression, with the regression equation and p-value reported at the top of each plot.

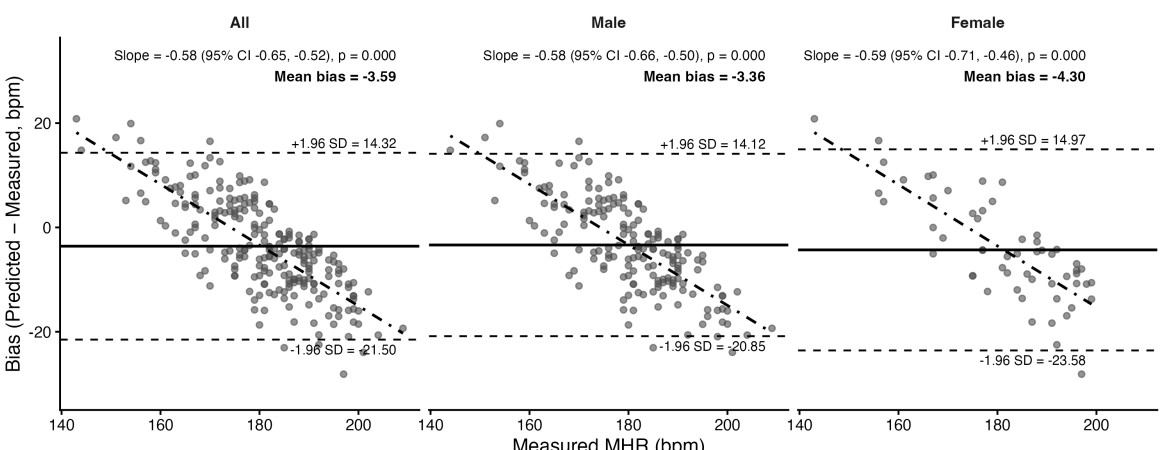

**Fig 17. Bland Altman analysis plots of maximum heart rate prediction for the Fairbairn equation.** Abbreviations: MHR = Maximum Heart Rate; bpm, beats per minute. The x-axis represents measured MHR in bpm, and the y-axis shows the difference between predicted and measured MHR (bias). The solid horizontal line represents the mean bias, while the upper and lower dashed lines indicate the 95% limits of agreement (LOA). The sloped dashed line shows the proportional bias estimated from linear regression, with the regression equation and p-value reported at the top of each plot.

## Discussion

The primary objective of this study was to determine whether CRF level influences predictive error across commonly used MHR prediction equations. The LMM revealed a significant interaction between prediction equation and VO₂max, indicating that the accuracy of MHR prediction equations was moderated by CRF level. Post-hoc analyses provided limited evidence of an overall CRF effect across the full sample; however, sex-stratified analyses demonstrated that this influence was more pronounced in males. For several equations (Fox, Tanaka, Arena, Åstrand, Nes), higher VO₂max was associated with greater prediction error, suggesting that fitter males tended to deviate more from their predicted MHR. Notably,

**Table 3. Agreement analysis results.**

| Equation | Sex | Mean bias | Proportional Bias (Yes/No, *p-value*) | MAE | RMSE | ICC |
|---|---|---|---|---|---|---|
| Fox | Both | 1.17 | No; *p*=0.616 | 8.44 | 10.28 | 0.64 |
| | Male | 1.36 | No; *p*=0.431 | 8.46 | 10.20 | 0.63 |
| | Female | 0.59 | No; *p*=0.800 | 8.38 | 10.52 | 0.67 |
| Gellish | Both | −0.29 | Yes; *p*<0.001 | 7.45 | 9.19 | 0.61 |
| | Male | −0.07 | Yes; *p*<0.001 | 7.33 | 8.99 | 0.60 |
| | Female | −0.97 | Yes; *p*<0.001 | 7.82 | 9.77 | 0.63 |
| Tanaka | Both | 0.71 | Yes; *p*<0.001 | 7.40 | 9.21 | 0.61 |
| | Male | 0.93 | Yes; *p*<0.001 | 7.32 | 9.04 | 0.60 |
| | Female | 0.03 | Yes; *p*<0.001 | 7.66 | 9.72 | 0.63 |
| Arena | Both | 1.24 | Yes; *p*<0.001 | 7.45 | 9.29 | 0.61 |
| | Male | 1.46 | Yes; *p*<0.001 | 7.40 | 9.14 | 0.60 |
| | Female | 0.56 | Yes; *p*=0.001 | 7.61 | 9.74 | 0.63 |
| Åstrand | Both | 3.93 | Yes; *p*=0.003 | 8.27 | 10.28 | 0.60 |
| | Male | 4.13 | Yes; *p*=0.031 | 8.30 | 10.23 | 0.59 |
| | Female | 3.29 | No; *p*=0.051 | 8.18 | 10.44 | 0.64 |
| Nes | Both | 6.02 | Yes; *p*<0.001 | 8.62 | 10.94 | 0.50 |
| | Male | 6.25 | Yes; *p*<0.001 | 8.73 | 10.89 | 0.49 |
| | Female | 5.31 | Yes; *p*<0.001 | 8.31 | 11.09 | 0.54 |
| Fairbairn | Both | −3.26 | Yes; *p*<0.001 | 8.19 | 9.92 | 0.59 |
| | Male | −2.93 | Yes; *p*=0.003 | 7.91 | 9.67 | 0.60 |
| | Female | −4.30 | Yes; *p*<0.001 | 9.07 | 10.66 | 0.56 |

Abbreviations: ICC, intraclass correlation coefficient; LOA, limits of agreement; MAE, mean absolute error; RMSE, root mean square error. Notes: ICC values were interpreted using the following thresholds: <0.50 = *poor*, 0.50–0.74 = *moderate*, 0.75–0.89 = *good*, and ≥0.90 = *excellent* reliability.

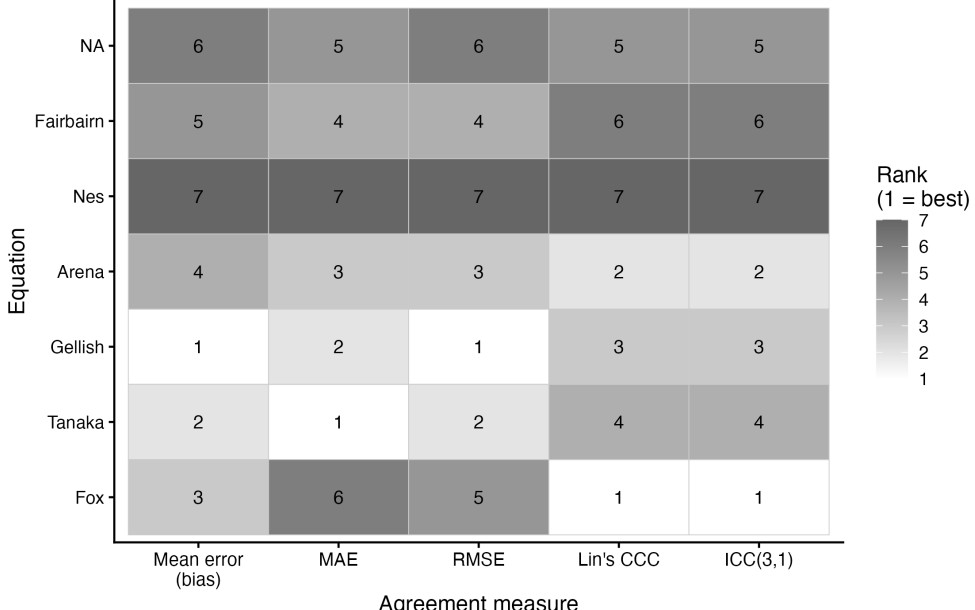

**Fig 18. Ranking of maximum heart rate prediction equations by accuracy metric for both sexes.** Abbreviations: CCC, Lin's concordance correlation coefficient; RMSE, root mean square error; MAE, mean absolute error; ICC, intraclass correlation coefficient. Bias is the mean bias from the Bland Altman analyses. The ranking indicates the performance of each equation per metric, with a lower ranking indicating better performance.

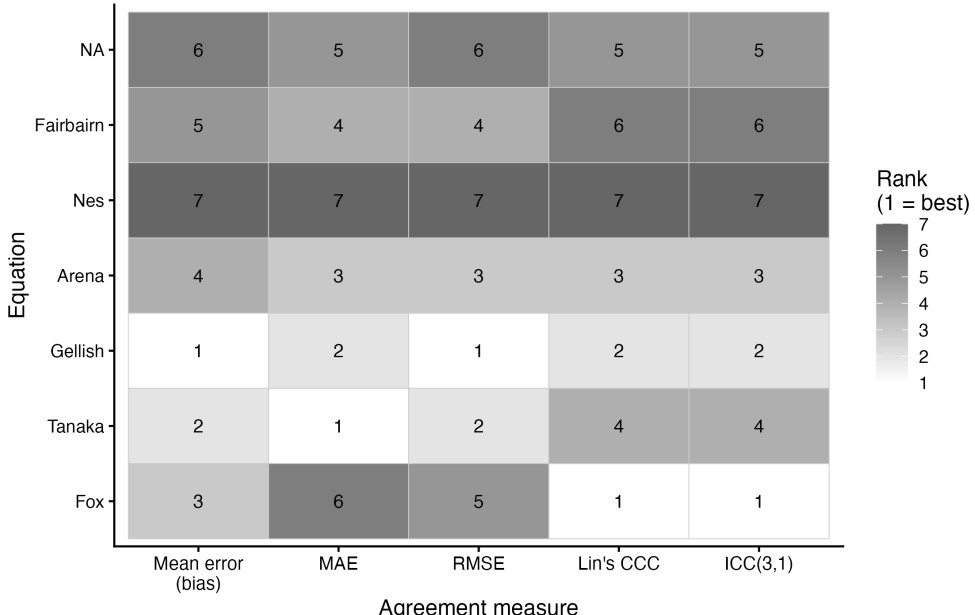

**Fig 19. Ranking of maximum heart rate prediction equations by accuracy metric for males.** Abbreviations: CCC, Lin's concordance correlation coefficient; RMSE, root mean square error; MAE, mean absolute error; ICC, intraclass correlation coefficient. Bias is the mean bias from the Bland Altman analyses. The ranking indicates the performance of each equation per metric, with a lower ranking indicating better performance.

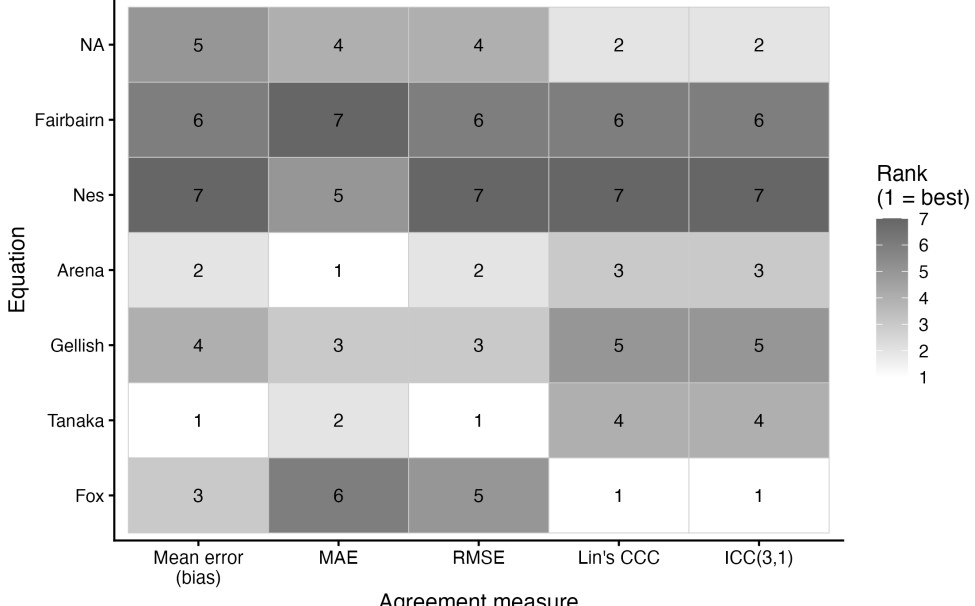

**Fig 20. Ranking of maximum heart rate prediction equations by accuracy metric for females.** Abbreviations: CCC, Lin's concordance correlation coefficient; RMSE, root mean square error; MAE, mean absolute error; ICC, intraclass correlation coefficient. Bias is the mean bias from the Bland Altman analyses. The ranking indicates the performance of each equation per metric, with a lower ranking indicating better performance.

the variance explained by VO$_2$max was small ($R^2 \leq 0.06$), highlighting that CRF had only a small influence on prediction accuracy. In contrast, females showed no meaningful relationship between CRF and prediction error, with consistently small and nonsignificant slopes across all equations.

Previous studies have reported inconsistent findings regarding the influence of aerobic training on MHR [16,17,19–21]. Some have proposed that endurance training may slightly reduce MHR due to autonomic adaptations and enhanced stroke volume [16,17,21], while others report negligible or variable effects depending on age, training history, or the duration of training and detraining [19,20]. More specifically, several studies have also reported minimal effects of CRF on MHR responses [1,22]. For example, Tanaka et al. [1] found that MHR is predominantly age-determined and largely independent of habitual physical activity (e.g., sedentary or exercise behaviors) leading to altered CRF [3]. Similarly, Lach et al. [22] reported that incorporating CRF and body composition into age-based MHR prediction models yielded only marginal improvements in model fit. The present study provides further evidence that CRF may have a weak influence on the accuracy of MHR prediction, particularly in males, with those who are fitter possibly having slightly great prediction errors. One possible explanation is that individuals with lower CRF may exhibit more uniform cardio-autonomic response (*i.e.*, limited stroke-volume reserve and typical sympathetic activation) at maximal exertion [32], resulting in MHR values that align more closely with age-based expectations. In contrast, higher CRF individuals may display greater physiological variability due to adaptations such as increased stroke volume, lower resting vagal set-point with different β-adrenergic sensitivity, and task-termination that may be influenced by peripheral factors, all of which could subtly affect MHR responses and increase prediction error [16,21,33].

A secondary aim of this study was to evaluate the overall accuracy of commonly used age-based MHR prediction equations stratified by sex and independent of CRF level. Bland–Altman analyses indicated that mean biases were generally small across equations, but wide limits of agreement (±18–24 bpm) highlighted substantial individual-level variability. Proportional bias was evident for most formulas, particularly Tanaka [1], Gellish [6], Arena [11], Nes [9], and Fairbairn [7], where predicted values increasingly underestimated measured MHR at higher heart rates. Notably, the Fox equation [4] showed neither significant mean nor proportional bias, reflecting relatively stable performance across the full range of MHR. When evaluating accuracy metrics, Tanaka [1] and Gellish [6] consistently demonstrated the lowest MAE and RMSE values, whereas Nes [9] produced the poorest agreement, with larger positive bias and significantly greater error compared with several other formulas. These patterns were largely consistent when stratified by sex, though females tended to show slightly wider limits of agreement and somewhat stronger proportional bias. Collectively, the findings support that although certain equations (e.g., Tanaka, Gellish, Arena) provide modestly better accuracy, all formulas exhibited considerable error margins, limiting their precision for individual-level application. These findings are consistent with prior work by Shookster et al. [13], who also found considerable individual error across MHR equations, proportional bias, and emphasized the limited utility of any single formula when applied universally.

## 4.1. Practical implications

One of the clearest implications of this study is the considerable individual variability in MHR, even after accounting for age, sex, and CRF. The 95% LOA spanned approximately ±20 bpm for all formulas. In practical terms, this degree of error could shift an individual across two heart rate training zones (e.g., from moderate to vigorous intensity), which has meaningful consequences for exercise prescription. Thus, while age-based MHR prediction equations are useful for providing population-level estimates and general fitness guidance (e.g., prescribing approximate target zones in a group class), they are not sufficiently precise for individualized programming, athletic performance optimization, or clinical decision-making. In these contexts, reliance on prediction equations could lead to systematic under- or over-prescription of training intensities, supporting the need for direct measurement of MHR when accuracy is critical.

For practitioners, clinicians, and researchers who rely on MHR for exercise prescription, screening, or monitoring, the results of this study highlight both the utility and the limitations of available equations. Among the prediction equations

evaluated, the Fox [4] formula exhibited arguably the most consistent performance across the sample, showing comparable error to Gellish [6], Tanaka [1], and Arena [11] while avoiding proportional bias. These characteristics support its continued use as a practical and generalizable option, especially in time-limited or equipment-limited settings. Given that all equations produced typical errors of approximately ±7–10 bpm, it is critical to treat age-based MHR predictions as approximate estimates rather than absolutes. Accordingly, exercise professionals should communicate this inherent uncertainty when prescribing training intensities based on predicted MHR. One approach is to present prediction intervals (*e.g.,* 175–185 bpm) instead of single-point estimates (*e.g.,* 180 bpm), thereby providing a more transparent representation of the expected range of error. Also, exercise professionals should consider additional methods, such as perceived exertion or heart rate monitoring over time with wearable devices [34], to refine exercise intensity for individuals. These approaches are supported by evidence highlighting both the limitations of heart rate–based prescriptions throughout an exercise session [35] and the value of RPE as a practical indicator of exercise intensity in applied contexts [36]. For instance, within-session cardiovascular drift and HR lag (*i.e.,* slow kinetics during rapid workload changes) can decouple heart rate from true metabolic load, limiting the precision of HR-based prescriptions [35]. In addition, medications such as β-blockers can blunt the chronotropic response [37], rendering HR-based zones misleading and necessitating greater reliance on alternative markers (*e.g.,* RPE, speed/power) in certain populations. Recent work also outlines treadmill protocols and monitoring strategies that improve constant-intensity prescription, such as step-ramp-step approaches in general and clinical populations [38] and individualized load management that integrates HR-based and subjective metrics [39]. Thus, incorporating multiple markers of exercise intensity can enhance safety, personalization, and client understanding when MHR is used for exercise prescription.

Finally, with respect to sex differences, our findings suggest that commonly used MHR prediction equations perform similarly for males and females once age is accounted for, with no meaningful systematic bias detected. Although females exhibited slightly greater variability, these differences were small and unlikely to warrant sex-specific prediction equations in healthy adult populations. This aligns with prior work by Tanaka et al. [1] and Nes et al. [9], who likewise concluded that sex does not substantially alter the age–MHR relationship. Nevertheless, the smaller female sample in our study limits our ability to detect subtle differences in variability or outlier behavior. Future work with larger and more balanced samples may help clarify whether nuanced sex-related distinctions exist. For practitioners, the key takeaway is that variability at the individual level remains the dominant source of error, outweighing sex-based differences.

### 4.2. Limitations

This study has several limitations. First, the sample was predominantly male (~76%), limiting the ability to fully generalize findings to females. Although we included the Fairbairn equation [7], which incorporates sex-specific coefficients, a more balanced sample would allow for stronger sex-based comparisons. Secondly, because this was a retrospective exploratory analysis of existing data, *a priori* power calculation was not conducted. As such, the study may be underpowered to detect small effects. Furthermore, underpowered design can also lead to an increase of the proportion of false positives in a field affected by publication bias [23]. However, from a practical perspective, large effects are of greater interest for practitioners, whereas small effects are unlikely to meaningfully influence training or clinical decision-making. Another potential limitation is the broad adult age range (18–68 y, including a subset >65 y) that may introduce heterogeneity (*e.g.,* age-related medications, chronotropic changes) that could attenuate associations [37,40]. Finally, while effort was made to ensure tests reached true physiological maximums with standard criteria [3], it is possible that some participants did not achieve their absolute MHR [26], introducing random error into the criterion measure. This may be in part due to the GXT protocol implemented in the present study. Although the protocol was designed for an ~8–12-minute time to task failure, using a grade-then-speed progression, graded running up to 10% incline may have increased local muscular discomfort, and caused some participants to terminate the test due to peripheral factors rather than central cardiorespiratory limitations. While $VO_2$max attainment criteria were met for all included tests, we cannot exclude the possibility that this underestimated true MHR in a subset of participants.

### 4.3. Future directions

Future researchers should continue to explore individual-level variability in MHR and seek models that improve prediction accuracy beyond age-based equations [1,4,6,7,9–11]. While adding CRF or body composition to regression models has produced only marginal gains [22], more complex methods, such as machine learning, may offer improvements when applied to large, diverse datasets [41]. Recent work using random forest models and other nonlinear approaches has shown potential to reduce prediction error by up to 20–25%, although individual-level variance remains high [41]. Further studies are also needed to evaluate MHR prediction accuracy in specific subpopulations, including older adults, individuals with cardiovascular or metabolic diseases, and elite endurance athletes, who may deviate from age-based norms due to factors like altered β-adrenergic responsiveness [33], medication effects [37], chronotropic incompetence [40], or training-induced cardiac remodeling that shifts the heart rate–work relationship [16].

## Conclusion

In summary, our findings highlight the substantial individual variability in MHR prediction, with small to no practically meaningful effect of CRF on prediction accuracy. Age-based MHR equations are useful for population-level benchmarking and for setting initial training targets, but their typical error ($\approx\pm7$–10 bpm) and inter-individual heterogeneity limit precision for individualized programming. Thus, when individual accuracy is needed, predicted MHR should be directly assessed and if possible complemented with additional markers (*e.g.,* RPE, speed, power). Future researchers should continue to explore individualized modeling approaches, though adjusting for CRF alone may not improve prediction accuracy among healthy adults. Importantly, the present results should be viewed as exploratory and confirmed in future confirmatory studies.

## Author contributions

**Conceptualization:** Joel Martin.

**Data curation:** Joel Martin, Courtney Gerrity.

**Formal analysis:** Joel Martin, Bryndan Lindsey, Courtney Gerrity.

**Funding acquisition:** Joel Martin.

**Investigation:** Joel Martin.

**Methodology:** Joel Martin, Bryndan Lindsey, Courtney Gerrity, Jatin Ambegaonkar.

**Project administration:** Joel Martin.

**Resources:** Joel Martin.

**Software:** Joel Martin.

**Supervision:** Joel Martin.

**Visualization:** Joel Martin, Bryndan Lindsey.

**Writing – original draft:** Joel Martin, Bryndan Lindsey, Courtney Gerrity, Jatin Ambegaonkar.

**Writing – review & editing:** Joel Martin, Bryndan Lindsey, Jatin Ambegaonkar.

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
