## [Decision Letter · Decision Letter 0]

6 Aug 2025

Dear Dr. Martin,

Thank you for submitting your manuscript to PLOS ONE. After careful consideration, we feel that it has merit but does not fully meet PLOS ONE’s publication criteria as it currently stands. Therefore, we invite you to submit a revised version of the manuscript that addresses the points raised during the review process.

We look forward to receiving your revised manuscript.

Kind regards,

Stefano Amatori, Ph.D.

Academic Editor

PLOS ONE

Journal Requirements:

2. We note that your Data Availability Statement is currently as follows: All relevant data are within the manuscript and in Supporting Information files.

Reviewers' comments:

Reviewer's Responses to Questions

**Comments to the Author**

1. Is the manuscript technically sound, and do the data support the conclusions?

Reviewer #1: Yes

Reviewer #2: Partly

Reviewer #3: Partly

2. Has the statistical analysis been performed appropriately and rigorously?

Reviewer #1: Yes

Reviewer #2: No

Reviewer #3: No

3. Have the authors made all data underlying the findings in their manuscript fully available?

Reviewer #1: Yes

Reviewer #2: Yes

Reviewer #3: Yes

4. Is the manuscript presented in an intelligible fashion and written in standard English?

Reviewer #1: Yes

Reviewer #2: Yes

Reviewer #3: Yes

Reviewer #1: Line 28: the years 2018-2023 are not the same as reported in the method section

Lines 71-73: you mention ‘numerous alternative equations’, you need to cite them year, preferably with the name of the equation and/or the author

Lines 88-89: add references for the mentioned studies/formulae

Lines 110-112: add a reference

Line 136: the years 2019-2024 are different to the years in the abstract

Discussion: state in the beginning of the discussion whether you can confirm your hypothesis or not

Reviewer #2: This manuscript, titled "Accuracy of Age-Based Maximal Heart Rate Equations in Adults with High and Low Cardiorespiratory Fitness Levels," aims to evaluate the predictive accuracy of seven common age-based MHR equations across different fitness levels. The study addresses an important and clinically relevant question. However, despite its clear structure and thorough data presentation, the work suffers from significant methodological limitations, inconsistent terminology, and analytical choices that reduce its scientific robustness. Below, I outline several major concerns that necessitate a substantial revision of the manuscript.

Abstract:

• Line 21: The acronym “MHR” is used here, while “HRmax” appears elsewhere. Please ensure consistency in terminology throughout the text.

• Line 37: In the abstract results section, p-values are reported. Given that p-values are highly dependent on sample size, it is recommended to also report effect sizes along with their confidence intervals.

Main Text:

• Line 134: It is essential to report the age distribution of the sample. A graphical representation (e.g., kernel density plot) would clarify whether the sample is approximately uniform or biased.

• Line 137: Sample size is a critical aspect of this type of research. In comparison to the studies cited in the introduction, the sample here appears underpowered. A power analysis is necessary to justify the chosen sample size, specifying the primary outcome and the minimum detectable effect.

• Line 142: Who were the participants? Why were they undergoing GXT testing? What population does this sample represent? These are crucial elements to support the study's external validity.

• Line 169: A scatterplot with age on the x-axis and HRmax on the y-axis should be included, with the regression lines for the seven equations superimposed.

• Line 182: Why was CRF treated as a binary variable? This dichotomization is not appropriate. All analyses involving CRF should be re-run treating it as a continuous covariate. Also, include a kernel density plot of CRF values, as well as a plot of CRF vs age, stratified by sex.

• Line 189: A repeated-measures ANOVA should be conducted using CRF as a covariate, and post hoc tests should compare the regression coefficients (R, a, and b) appropriately.

• Line 198: All analyses should be repeated using CRF as a continuous covariate.

• Line 236: The dichotomization into “high” and “low” CRF groups is arbitrary and not physiologically grounded. Eliminate this split and instead stratify by sex.

• The term “female” should be removed from the table where not necessary.

• Cohen’s d values should be reported as absolute values.

• Figure 1: Some predicted MHR values are unreasonably low and likely reflect data entry or estimation errors; these points should be removed. Since all prediction methods are linear equations, distributions should vary only by intercept and scale, not shape. This consistency is not reflected in the figure, indicating possible issues in its construction.

While the topic of this study is undoubtedly relevant and the manuscript is generally well-written, the current version suffers from multiple critical issues that compromise its scientific validity:

• The sample appears underpowered and lacks justification through a power analysis.

• CRF is inappropriately treated as a categorical variable, introducing artificial thresholds and reducing statistical power.

• Key descriptive data (e.g., age and CRF distributions) are missing.

• Graphical representations are either lacking or flawed.

• The statistical methods, particularly in regard to interaction effects and covariate inclusion, require complete re-evaluation.

Given these major concerns, I do not believe the manuscript is ready for publication in its current form. However, with a thorough methodological revision, clearer data visualization, and a more nuanced statistical approach (especially using CRF as a continuous covariate), the manuscript could be resubmitted and reevaluated. Recommendation: Major Revision

Reviewer #3: General comments:

In general, the article is well written, and the authors have made an effort to be as precise as possible in respecting the guidelines (e.g., providing information about data access, etc.). However, the study requires improvements in certain aspects of writing and a better justification for the use of specific statistical approaches. Before delving deeper into the specific comments, it is recommended to revise (and add line numbers, as well as after Table 2) the manuscript based on the significant points outlined below.

Title:

As this is a retrospective study, it would be advisable to classify and describe it as exploratory. Including the word exploratory in the title could be helpful. Please refer to the recent article by Ditroilo and colleagues (PMID: 40197233) on the importance of exploratory research (see detailed comments below).

Abstract:

The abstract, in general, should be reformed once the changes are made in the main text.

Introduction:

While the Introduction is well written, it is slightly too long. It would be more effective if it were more concise and ideally no longer than 1.5 to 2 pages. Please reduce the content.

Specific lines comments:

Line 82: The sentence “Furthermore, sex can influence cardiovascular responses to exercise” feels out of context. Consider providing a better transition or justification.

Lines 127–128: Avoid stating hypotheses, especially in retrospective studies where a priori hypothesis testing is not feasible and could lead to the so-called HARKing (see Mesquida et al., 2022; PMID: 36533197). Limit this section to stating the main objectives. Again, the article by Ditroilo et al. (PMID: 40197233) may help in distinguishing between exploratory and confirmatory research.

Materials and methods:

The number of the ethical approval should also be present in the main text.

Statistics:

Please specify how data distribution was assessed, and which tests were used for non-normally distributed data, if applicable.

Regarding sample size: since this is a retrospective study, there may be no formal justification. If that is the case, it is still important to disclose this information.

Please provide a justification for the use of a dichotomous classification of CRF (High vs. Low) based on the 50th percentile. Dichotomizing a continuous variable such as VO₂max may lead to the loss of important information. If possible, consider treating CRF as a continuous variable. On the contrary, if it is not possible for whatever reason, try to justify the decision.

Specific lines comments:

Lines 150–151: The sentence “After a brief warm-up, the treadmill protocol ramped intensity continuously via speed and/or grade increments every 1–2 minutes” is too vague. Please provide a more detailed description of the protocol.

Lines 180-182: Please, see the comment above on the division between high and low fitness levels.

Lines 209-212: It is not clear how the top and bottom were determined. In particular, a justification of the criteria for this decision should be made.

Line 211: Use a subscript for VO₂max.

Results:

After Table 2, line numbers are missing. Please correct this.

The choice of the 20th percentile at the top and the bottom remains unjustified and seems arbitrary.

Discussion:

Line numbers are missing, making it difficult to reference specific points. Please address this.

In general, the discussion should be reframed within the context of exploratory research. For instance, hypotheses should be proposed for future confirmatory studies rather than being inferred from the current findings. Moreover, a part related to the presence of interindividual variability should be incorporated and discussed.

• From the sentence “This suggests that although individuals with lower CRF may exhibit slightly smaller prediction errors...” to the end of the paragraph should be revised in light of the exploratory nature of the study.

• Figure 3 should be moved from the discussion to the results section.

• The sentence “Although there were no statistically significant differences in prediction accuracy between High and Low CRF individuals in the primary analysis, the descriptive analyses revealed that individuals with higher CRF may exhibit greater prediction error across most equations” is particularly important. It is commendable that non-significant results are reported and discussed as did in the present study! Is is also possible to push this aspect in your discussion.

• The statement “Exercise professionals should consider additional methods, such as perceived exertion or heart rate monitoring over time with wearables” is valid and could be expanded. Consider referencing Teso et al. (2022; PMID: 34967799) and Ferri Marini et al. (2024; PMID: 38711624) for additional context on heart rate prescription limitations and the relationship between RPE and physiological measures.

• The sentence “Specifically, rather than offering a single-point estimate (e.g., “Your MHR is 180 bpm”), professionals can provide prediction intervals (e.g., “Your MHR is likely between 175–185 bpm”) to aid their clients’ understanding of the errors associated with MHR estimation.” sounds colloquial, it can be reformed in a more scientific way.

• The sentence “Alternatively, when precision is critical, such as in clinical exercise testing or high-performance athletics, direct MHR measurement via graded exercise testing remains the gold standard” is debatable. In high-performance settings, maximal testing may not be feasible due to potential training disruption. Consider revising or removing this statement.

Conclusion:

It is important to state in the conclusion that all the findings of the present study should be confirmed in future confirmatory studies.

Figures:

Please, check all the figure titles and labels. For instance, Figure 2 (a, b, c, etc.) contains a spelling error in the left panel (i.e., Cardorespiratory fitness instead of Cardiorespiratory). Still, in the figures, although they appear clear and sharp when downloaded, they are somewhat blurred in the PDF file. Verify that they are meeting the minimum quality requirements and, if necessary, adjust them accordingly.

Data availability:

Even if the data are uploaded correctly to OSF, it is still necessary to cite them in the main text. Furthermore, it is probably better to anonymize the data.

**Do you want your identity to be public for this peer review?** For information about this choice, including consent withdrawal, please see our Privacy Policy

Reviewer #1: No

Reviewer #2: No

Reviewer #3: No

---

## [Author Response · Author response to Decision Letter 1]

4 Sep 2025

PLOS ONE Editor(s) and reviewers,

We are pleased to re-submit the following manuscript now titled: “Exploratory Analysis of the Accuracy of Age-Based Maximal Heart Rate Equations Across Cardiorespiratory Fitness Levels”. With our resubmission we are providing a revised manuscript with changes noted in track changes and this response letter. Point by point responses to comments are provided below. We would like to express our appreciation to each of the reviewers and the editor for their time.

Reviewer #1

Line 28: the years 2018-2023 are not the same as reported in the method section

Response: Thank you for bringing this to our attention. The years have been corrected. The correct years are 2019-2024.

Lines 71-73: you mention ‘numerous alternative equations’, you need to cite them year, preferably with the name of the equation and/or the author

Response: We appreciate the comment. We’ve added citations to the sentence noted in the review. This sentence is intended to transition to the subsequent paragraph which describes equations in more detail with the equation name.

Lines 88-89: add references for the mentioned studies/formulae

Response: Thank you for the comment. References have been added.

Lines 110-112: add a reference

Response: We have deleted this sentence altogether. Another reviewer suggested that the introduction be shorten and as part of that revision this sentence was decided to be removed.

Line 136: the years 2019-2024 are different to the years in the abstract

Response: The years have been corrected in the abstract.

Discussion: state in the beginning of the discussion whether you can confirm your hypothesis or not.

Response: We appreciate this suggestion. However, in response to another reviewer’s feedback we revised the manuscript to emphasize that this study is exploratory in nature and does not test a priori hypotheses. Instead, we framed our work around the stated objectives, which focus on evaluating the accuracy of multiple MHR prediction equations across CRF levels. Accordingly, the discussion now begins with a summary of the main findings relative to these objectives, rather than a confirmation or rejection of hypotheses. We believe this aligns with current best practices for reporting exploratory research described by Ditroilo et al.

Ditroilo M, Mesquida C, Abt G, Lakens D. Exploratory research in sport and exercise science: Perceptions, challenges, and recommendations. Journal of Sports Sciences. 2025;43: 1108–1120. doi:10.1080/02640414.2025.2486871

Reviewer #2

This manuscript, titled "Accuracy of Age-Based Maximal Heart Rate Equations in Adults with High and Low Cardiorespiratory Fitness Levels," aims to evaluate the predictive accuracy of seven common age-based MHR equations across different fitness levels. The study addresses an important and clinically relevant question. However, despite its clear structure and thorough data presentation, the work suffers from significant methodological limitations, inconsistent terminology, and analytical choices that reduce its scientific robustness. Below, I outline several major concerns that necessitate a substantial revision of the manuscript.

Abstract:

• Line 21: The acronym “MHR” is used here, while “HRmax” appears elsewhere. Please ensure consistency in terminology throughout the text.

Response: Thank you for bring this to our attention. We have proofread to make sure only MHR is used in the text.

• Line 37: In the abstract results section, p-values are reported. Given that p-values are highly dependent on sample size, it is recommended to also report effect sizes along with their confidence intervals.

Response: We agree with the reviewer’s comment regarding the importance of reporting effect sizes alongside p-values. Due to substantial revisions to our analyses, the p-values originally reported in the abstract are no longer present in the revised version. The current abstract highlights variance explained (marginal and conditional R² from LME models) and regression coefficients with R² values where appropriate, as these provide interpretable indicators of effect magnitude. While traditional standardized effect sizes with confidence intervals were not available for all models (e.g., LME), we made every effort to present complementary measures of model fit and accuracy (e.g., MAE, RMSE, ICC) that contextualize the results beyond statistical significance throughout the manuscript.

Main Text:

• Line 134: It is essential to report the age distribution of the sample. A graphical representation (e.g., kernel density plot) would clarify whether the sample is approximately uniform or biased.

Response: We appreciate the reviewer’s suggestion. In the revised manuscript we have added a detailed description of the age distribution of the sample for the overall cohort. To further illustrate the distribution, we have also included a kernel density plot (new Figure 1) that clearly depicts the age distribution across participants. Results of the Shapiro–Wilk test indicated that age was not normally distributed, with the sample showing a greater proportion of younger participants and a smaller number of older participants (right-skewed distribution). These additions allow readers to both quantitatively and visually assess the distribution of age in the sample.

• Line 137: Sample size is a critical aspect of this type of research. In comparison to the studies cited in the introduction, the sample here appears underpowered. A power analysis is necessary to justify the chosen sample size, specifying the primary outcome and the minimum detectable effect.

Response: We thank the reviewer for this important comment. Because this was a retrospective analysis of graded exercise test data, no a priori power calculation was conducted. Instead, all available data that met inclusion criteria were included. We recognize that this limits our ability to make definitive claims about minimum detectable effects. To address this, we have clarified in the Participants and Design section that the study is exploratory in nature. This framing aligns with recent recommendations for exploratory research in sport and exercise science (Ditroilo et al.), which emphasize transparency and hypothesis-generating insights rather than confirmatory, power-based justifications. We believe this approach is appropriate for the current dataset and clearly communicates the scope and limitations of our findings.

Ditroilo M, Mesquida C, Abt G, Lakens D. Exploratory research in sport and exercise science: Perceptions, challenges, and recommendations. Journal of Sports Sciences. 2025;43: 1108–1120. doi:10.1080/02640414.2025.2486871

• Line 142: Who were the participants? Why were they undergoing GXT testing? What population does this sample represent? These are crucial elements to support the study's external validity.

Response: We thank the reviewer for this important comment. We have revised the Participants and Design section to clarify that participants were individuals from the local community who voluntarily accessed our laboratory’s human performance testing services. Most were recreational athletes of varying levels seeking VO₂max and MHR data for training purposes, while others underwent testing for general health assessment. These details have been incorporated to better describe the study population and improve the interpretation of external validity (see Section 2.1).

• Line 169: A scatterplot with age on the x-axis and HRmax on the y-axis should be included, with the regression lines for the seven equations superimposed.

Response: We agree and have added a scatterplot of measured MHR versus age with superimposed regression lines for the seven age-based prediction equations, including sex-specific lines for the Fairbairn model (new Figure 2). This visualization enables direct comparison between observed values and each equation across the age range.

• Line 182: Why was CRF treated as a binary variable? This dichotomization is not appropriate. All analyses involving CRF should be re-run treating it as a continuous covariate. Also, include a kernel density plot of CRF values, as well as a plot of CRF vs age, stratified by sex.

Response: We have re-run all analyses treating CRF (VO₂max) as a continuous covariate rather than a binary classification. In addition, we now include (i) a kernel density plot of VO₂max to illustrate its distribution (new Figure 1C) and (ii) a scatterplot of VO₂max versus age stratified by sex (new Figure 2). Additionlly, we tested whether the slope of the VO2max vs. age relationship differs for men and women. We found no difference and report this in the results as part of preliminary analyses. These additions address the reviewer’s requests and enhance the transparency of the CRF distribution and its relationship with age.

• Line 189: A repeated-measures ANOVA should be conducted using CRF as a covariate, and post hoc tests should compare the regression coefficients (R, a, and b) appropriately.

Response: We thank the reviewer for this suggestion. In the revised manuscript, we used linear mixed-effects models (LMMs) to address the reviewer’s concern. Rather than categorizing CRF, we modeled VO₂max as a mean-centered continuous covariate. This approach avoids the information loss associated with grouping or dichotomizing CRF and allows direct estimation of how prediction error changes across the full spectrum of fitness. Because each subject contributed multiple predictions (across seven equations), the independence assumption of ANOVA was not met. By including a random intercept for participant, the LMM explicitly accounts for within-subject correlation, which a standard ANCOVA cannot do without restrictive assumptions. We also included the Equation × VO₂max interaction to test whether CRF modified equation-specific accuracy.

To address the reviewer’s request for post hoc tests, we extracted estimated marginal means and simple slopes from the LMMs using the emmeans package, with pairwise contrasts adjusted by the Tukey method. This provides equivalent interpretive value to ANOVA post hoc tests, while accommodating unequal variances, unbalanced data, and random effects.

To further quantify the role of CRF, we fit separate OLS regression models for each equation, regressing absolute error on VO₂max. These models provided slopes (with 95% CI, p-values, R², and RMSE) that quantify whether greater CRF systematically increased or decreased error. The plots in Figure 4 illustrate these equation-specific slopes, stratified by sex.

Taken together, the LMMs evaluate the fixed effects of Equation, VO₂max, and their interaction while modeling the correlated data structure, and the regression slopes provide equation-specific estimates of CRF’s modifying effect. We believe this approach appropriately addresses the reviewer’s concern and strengthens the analysis by including CRF as a continuous covariate but are open to additional feedback to improve the manuscript.

• Line 198: All analyses should be repeated using CRF as a continuous covariate.

Response: Thank you for the feedback. This was addressed as described in a response above.

• Line 236: The dichotomization into “high” and “low” CRF groups is arbitrary and not physiologically grounded. Eliminate this split and instead stratify by sex.

Response: Thank you for the feedback. This was addressed as described in a response above.

• The term “female” should be removed from the table where not necessary.

Response: The term “female” has been removed from the tables. When refering to sex we now more use “men” and “women” throughout.

• Cohen’s d values should be reported as absolute values.

Response: Thank you for the feedback. This has been corrected in the revised manuscript.

• Figure 1: Some predicted MHR values are unreasonably low and likely reflect data entry or estimation errors; these points should be removed. Since all prediction methods are linear equations, distributions should vary only by intercept and scale, not shape. This consistency is not reflected in the figure, indicating possible issues in its construction.

Response: Thank you for bringing this to our attention. Upon further review of the data there were 3 participants whom the integrity of their data appears low based on their age and MHR. We formally now removed outliers using prediction errors greater than 3 standard deviations. This led to the 3 participants being removed and our sample is now 230. We’ve also added figure 2 (MHR vs. age) to illustrate the linear relationship of the prediction equations with age. We appreciate the feedback on this matter.

While the topic of this study is undoubtedly relevant and the manuscript is generally well-written, the current version suffers from multiple critical issues that compromise its scientific validity:

• The sample appears underpowered and lacks justification through a power analysis.

Response: We thank the reviewer for raising these important points. Regarding statistical power, we acknowledge that no a priori power analysis was conducted. Because this was a retrospective analysis of an existing dataset, the sample size was determined by data availability rather than by an a priori calculation. We have clarified this in the revised manuscript (Introduction and Participants sections). We also explicitly note that this work should be considered exploratory in nature, consistent with recent recommendations to distinguish exploratory from confirmatory research. While the sample may be underpowered for certain effects, the purpose of this study is to generate preliminary insights into the accuracy of commonly used MHR equations across different levels of CRF, which can guide future confirmatory work with larger, prospectively designed samples. In addition, we have revised our analytical approach so that CRF is no longer treated categorically. Instead, VO₂max was modeled as a continuous covariate in linear mixed-effects models. This avoids introducing artificial thresholds, reduces loss of statistical power, and allows for more nuanced evaluation of how CRF influences prediction error across the full spectrum of fitness levels.

This being said the reviewer’s points are noted and have been added as potential limitations as well.

We believe these revisions strengthen the methodological rigor and address the reviewer’s concerns about both statistical validity and the treatment of CRF in the analysis.

• CRF is inappropriately treated as a categorical variable, introducing artificial thresholds and reducing statistical power.

Response: We appreciate the comment and have responded to similar comments above.

• Key descriptive data (e.g., age and CRF distributions) are missing.

Response: Thank you for the feedback. These data have been added.

• Graphical representations are either lacking or flawed.

Response: All the figures have been either replaced, updated or removed per the earlier feedback from the reviewer.

• The statistical methods, particularly in regard to interaction effects and covariate inclusion, require complete re-evaluation.

Response: The statistical analysis has been completely redone per the other points of feedback from the reviewer.

Given these major concerns, I do not believe the manuscript is ready for publication in its current form. However, with a thorough methodological revision, clearer data visualization, and a more nuanced statistical approach (especially using CRF as a continuous covariate), the manuscript could be resubmitted and reevaluated. Recommendation: Major Revision

Response: We thank the reviewer for their careful evaluation and constructive feedback. In response, we have thoroughly revised the manuscript to address the methodological, statistical, and visualization concerns. Specifically, we implemented a more nuanced statistical approach by incorporating CRF as a continuous covariate within linear mixed-effects models, which properly accounts for within-subject correlation and avoids the information loss associated with categorizing CRF. To complement these models, we also fit separate regression analyses for each equation to provide equation-specific estimates of the effect of CRF on absolute prediction error

---

## [Decision Letter · Decision Letter 1]

6 Oct 2025

Dear Dr. Martin,

Thank you for submitting your manuscript to PLOS ONE. After careful consideration, we feel that it has merit but does not fully meet PLOS ONE’s publication criteria as it currently stands. Reviewer #3 asked for some additional minor changes that could help to strengthen the manuscript further. Therefore, we invite you to submit a revised version of the manuscript that addresses the points raised during the review process. 

We look forward to receiving your revised manuscript.

Kind regards,

Stefano Amatori, Ph.D.

Academic Editor

PLOS ONE

Journal Requirements:

Reviewers' comments:

Reviewer's Responses to Questions

**Comments to the Author**

Reviewer #1: All comments have been addressed

Reviewer #3: (No Response)

2. Is the manuscript technically sound, and do the data support the conclusions?

Reviewer #1: Yes

Reviewer #3: Yes

3. Has the statistical analysis been performed appropriately and rigorously?

Reviewer #1: Yes

Reviewer #3: Yes

4. Have the authors made all data underlying the findings in their manuscript fully available?

Reviewer #1: Yes

Reviewer #3: Yes

5. Is the manuscript presented in an intelligible fashion and written in standard English?

Reviewer #1: Yes

Reviewer #3: Yes

Reviewer #1: The authors have addressed all comments. No further changes are needed. The submission is ready to be accepted for publication

Reviewer #3: General comments:

Overall, the authors have made a commendable effort to improve the manuscript, implementing appropriate modifications where necessary. Below are some specific comments that may help further enhance the quality and readability of the work.

Specific comments:

Line 65. The sentence “which can be impractical and/or risky for many” can be improved and given in a more scientific language.

Line 82. The sentence “sex can also influence cardiovascular responses to exercise”, while can be generally understood, contains a fundamental pitfall. Indeed, it is common for researchers to refer to sex as something that can influence responses to exercise, but this interpretation seems to suggest a reverse causality. This sentence should be then rephrased giving something like sexes can respond differently to exercise.

Line 106. It could be avoided to define age as a “dominant predictor” if considering that usually the predictor is defined as something that shows whether something is likely to happen.

Lines 137-139. The sample size part is now well explained, the recommendation here is also to cite the article from Lakens (https://doi.org/10.1525/collabra.33267) on the sample size justification and adapt the sentence consequently.

Lines 195-196. For this procedure does not seem an arbitrary decision, please provide a citation here.

Line 218. There is one more space after “interaction”, please remove it and check all over the text if this happened again.

GXT testing protocol. Then, all the participants experienced (about) the same starting speed and the same increments? If this is the case, was the time to task failure between 8 and 12 for every participant? Moreover, how running in a 10% slope could have influenced the responses to exercise? Being a not common way of running it could be expected that people were not able to reach their maximum effort because of leg pain or other peripheral discomfort. The impact should be at least acknowledged.

Table 1. This table can be improved visually. Indeed, it is advisable not to use all the borders but just the one at the top (i.e., below headings) and one at the bottom (i.e., between the last row and the notes). Additionally, it is suggested to use males and females instead of men and women, both in this table and all over the text.

Table 2 is formatted as suggested for Table 1. It is advisable to be consistent across all the tables and adjust also Table 3 etc.

Lines 412-413. Please, use males and females instead of men and women.

Discussion. While the range of age is quite wide in the present study, there is no mention of it as a possible limitation. It should be included, at least discussed as the age ranges also outside the usual definition of adults (e.g., also people above 65 yrs old). How could it have an impact on the results?

Line 425. Please, modify men and women also in the discussion.

Line 444. The sentence “may exhibit more stereotypical physiological responses to maximal exertion” should be improved and explained more to allow the reader to understand the meaning of this paragraph.

Lines 493-495. These lines are correct and explain a good point. The only suggestion here could be to explain just in a few more lines why HR has limitations (i.e., the presence of a cardiovascular drift).

Lines 495-497. In this line there is a great point. For instance, regarding safety, there always should be awareness on the fact that in certain situations where HR cannot be used or can be a misleading parameter (e.g., patients under beta-blockers medications) require attention to different parameters for training monitoring. To be general, you can also expand this section using references of papers that tried to prescribe and monitor aerobic exercise intensity in treadmill running using different approaches and parameters, such as the papers in both general and clinical populations from Faricier et al (PMID: 38980336 and PMID: 39999365) and the papers from Nuuttila et al (PMID: 38956784 and PMID: 35975912). These references and topics could add great value to this piece of discussion.

Lines 515-516. The sentence “As such, the study may be underpowered to detect small effects.” is just partially correct. Indeed, medium effects can be difficult to get with an underpowered design. Furthermore, underpowered design can also lead to an increase of the proportion of false positives in a literature where publication bias is present. It is advisable to include both in the text. For a reference on that, please check the article from Mesquida et al. (PMID: 36533197), in particular in section 2.2.2 of the paper (i.e., “Consequences of underpowered designs”).

Lines 531-532. The sentence “who may systematically deviate from general trends.” Should be expanded in the view of making readers understand what is meant.

Lines 536-538. The sentence “The results illustrate both the utility and the limitations of age-based MHR equations—useful for population-level estimates, yet insufficiently precise for individualized programming.” is to some extent counterintuitive. Indeed, it should be better explained why something could show both utility and limitations. Probably this sentence is more suitable for a discussion section, instead of a conclusion.

Figures. Please, as already asked for in the text, make sure that in all figures men and women are changed with males and females.

Figures 5 A, B, C, D, E, F, G. Please, try to avoid that the mean bias is overlapped with the plot dots.

OSF raw data file. Please, consider modifying the column name which refers to gender. Indeed, it should not be confused with sex, which is more related to what has been made in the present article (i.e., it was considered as a biological factor).

**Do you want your identity to be public for this peer review?** For information about this choice, including consent withdrawal, please see our Privacy Policy

Reviewer #1: No

Reviewer #3: No

---

## [Author Response · Author response to Decision Letter 2]

10 Oct 2025

PLOS ONE Editor(s) and reviewers,

We are pleased to re-submit the following manuscript: “Exploratory Analysis of the Accuracy of Age-Based Maximal Heart Rate Equations Across Cardiorespiratory Fitness Levels”. With our second resubmission we are providing a revised manuscript with changes noted in track changes and this response letter. Point by point responses to comments are provided below. We would like to express our appreciation to each of the reviewers and the editor for their time.

Reviewer #1

The authors have addressed all comments. No further changes are needed. The submission is ready to be accepted for publication

Response: Thank you for your positive feedback and recommendation. We appreciate your time and support throughout the review process.

Reviewer #2

Overall, the authors have made a commendable effort to improve the manuscript, implementing appropriate modifications where necessary. Below are some specific comments that may help further enhance the quality and readability of the work.

Response: Thank you for your positive feedback and recommendation. We are especially grateful for your thoughtful suggestions during the first round of review, which helped us significantly improve the quality of the paper. We appreciate your time and support throughout the process.

Specific comments:

Line 65. The sentence “which can be impractical and/or risky for many” can be improved and given in a more scientific language.

Response: Thank you for the feedback. We agree that the writing was colloquial and have revised to better reflect scientific language and improve clarity.

Line 82. The sentence “sex can also influence cardiovascular responses to exercise”, while can be generally understood, contains a fundamental pitfall. Indeed, it is common for researchers to refer to sex as something that can influence responses to exercise, but this interpretation seems to suggest a reverse causality. This sentence should be then rephrased giving something like sexes can respond differently to exercise.

Response: Thank you for pointing out this important nuance. We agree that the original phrasing may have unintentionally suggested reverse causality. We have revised the sentence to: “While these MHR prediction equations rely primarily on age, cardiovascular responses to exercise may differ between sexes due to physiological and hormonal differences [12],” to more accurately reflect the relationship and improve scientific clarity.

Line 106. It could be avoided to define age as a “dominant predictor” if considering that usually the predictor is defined as something that shows whether something is likely to happen.

Response: Thank you for pointing this out. We agree that calling age a “dominant predictor” could imply a causal or probabilistic role beyond what our data and prior literature support. Our intent was to note that age is typically the most informative single covariate in population equations for MHR, not to imply determinism.

Lines 137-139. The sample size part is now well explained, the recommendation here is also to cite the article from Lakens (https://doi.org/10.1525/collabra.33267) on the sample size justification and adapt the sentence consequently.

Response: We appreciate the suggestion to cite Lakens. We revised the Methods to (i) cite Lakens’ framework on sample-size justification and (ii) explicitly justify our approach for a retrospective exploratory analysis. We also clarify that future confirmatory work should plan sample sizes for desired precision or a smallest effect size of interest.

Lines 195-196. For this procedure does not seem an arbitrary decision, please provide a citation here.

Response: Thank you for the comment. We added some text to explain the decision for the outlier detection procedure.

Line 218. There is one more space after “interaction”, please remove it and check all over the text if this happened again.

Response: Thank you for the edit. We fixed in this line and proofread the rest of the document to correct any other instances.

GXT testing protocol. Then, all the participants experienced (about) the same starting speed and the same increments? If this is the case, was the time to task failure between 8 and 12 for every participant? Moreover, how running in a 10% slope could have influenced the responses to exercise? Being a not common way of running it could be expected that people were not able to reach their maximum effort because of leg pain or other peripheral discomfort. The impact should be at least acknowledged.

Response: Thank you for this helpful comment. In our dataset, time to task failure was approximately 8–12 minutes for the majority of tests (~≥80%), although exact durations were not available for all records. Starting speed varied by running ability, and it was uncommon for participants to reach the 10% grade. We agree that running at a 10% slope can increase local muscular demand and may prompt earlier termination due to peripheral discomfort rather than cardiorespiratory limitation. We have clarified these points in the Methods (individualized starting speed, grade-then-speed progression, 8–12-minute target) and added a Limitation noting that use of a 10% grade could, in some cases, constrain attainment of true maximal effort and thereby slightly underestimate MHR in a subset of participants.

Table 1. This table can be improved visually. Indeed, it is advisable not to use all the borders but just the one at the top (i.e., below headings) and one at the bottom (i.e., between the last row and the notes). Additionally, it is suggested to use males and females instead of men and women, both in this table and all over the text.

Response: Thank you for the feedback. The changes have been made to tables 1 and 3.

Table 2 is formatted as suggested for Table 1. It is advisable to be consistent across all the tables and adjust also Table 3 etc.

Response: Thank you again for the feedback. We have revised tables 1 and 3 accordingly.

Lines 412-413. Please, use males and females instead of men and women.

Response: We appreciate the reviewer bringing this to our attention. The edit has been made throughout the manuscript.

Discussion. While the range of age is quite wide in the present study, there is no mention of it as a possible limitation. It should be included, at least discussed as the age ranges also outside the usual definition of adults (e.g., also people above 65 yrs old). How could it have an impact on the results?

Response: Thank you for this suggestion. We added a limitation noting that our broad adult age range (18–68 y, including a subset >65 y) may introduce heterogeneity via age-related medications and chronotropic changes that could attenuate associations.

Line 425. Please, modify men and women also in the discussion.

Response: We appreciate the reviewer bringing this to our attention. The edit has been made throughout the manuscript.

Line 444. The sentence “may exhibit more stereotypical physiological responses to maximal exertion” should be improved and explained more to allow the reader to understand the meaning of this paragraph.

Response: Thank you for this suggestion. We revised the sentence to avoid the vague term “stereotypical” and to explain the proposed mechanism more explicitly.

Lines 493-495. These lines are correct and explain a good point. The only suggestion here could be to explain just in a few more lines why HR has limitations (i.e., the presence of a cardiovascular drift).

Response: Thank you for the suggestion. We added a sentence explaining why HR can be limited for within-session prescription, emphasizing cardiovascular drift and lag as common challenges.

Lines 495-497. In this line there is a great point. For instance, regarding safety, there always should be awareness on the fact that in certain situations where HR cannot be used or can be a misleading parameter (e.g., patients under beta-blockers medications) require attention to different parameters for training monitoring. To be general, you can also expand this section using references of papers that tried to prescribe and monitor aerobic exercise intensity in treadmill running using different approaches and parameters, such as the papers in both general and clinical populations from Faricier et al (PMID: 38980336 and PMID: 39999365) and the papers from Nuuttila et al (PMID: 38956784 and PMID: 35975912). These references and topics could add great value to this piece of discussion.

Response: Thank you. We expanded the final sentences plus cited examples of alternative/intensified monitoring and prescription approaches in both general and clinical populations. We appreciate the Faricier et al and Nuuttila et al. articles shared.

Lines 515-516. The sentence “As such, the study may be underpowered to detect small effects.” is just partially correct. Indeed, medium effects can be difficult to get with an underpowered design. Furthermore, underpowered design can also lead to an increase of the proportion of false positives in a literature where publication bias is present. It is advisable to include both in the text. For a reference on that, please check the article from Mesquida et al. (PMID: 36533197), in particular in section 2.2.2 of the paper (i.e., “Consequences of underpowered designs”).

Response: Thank you for this suggestion. We now clarify that limited power can impede detection of small to medium effects and, in literatures with publication bias, may inflate the proportion of false positives. We added Mesquida et al. as a reference and updated the Limitations accordingly.

Lines 531-532. The sentence “who may systematically deviate from general trends.” Should be expanded in the view of making readers understand what is meant.

Response: Thank you. We expanded the sentence to specify how these subgroups may differ (medications, chronotropic changes, training adaptations), clarifying why individual prediction error can be systematically biased in these populations.

Lines 536-538. The sentence “The results illustrate both the utility and the limitations of age-based MHR equations—useful for population-level estimates, yet insufficiently precise for individualized programming.” is to some extent counterintuitive. Indeed, it should be better explained why something could show both utility and limitations. Probably this sentence is more suitable for a discussion section, instead of a conclusion.

Response: Thank you for noting the potential ambiguity. We revised the language to explicitly state when age-based MHR equations are useful (group-level/initial targets) and when they are limited (individualized programming given typical ±7–10 bpm error and heterogeneity).

Figures. Please, as already asked for in the text, make sure that in all figures men and women are changed with males and females.

Response: This change has been made to all figures.

Figures 5 A, B, C, D, E, F, G. Please, try to avoid that the mean bias is overlapped with the plot dots.

Response: Thank you for this suggestion. We have moved the mean bias label to the top of each plot.

OSF raw data file. Please, consider modifying the column name which refers to gender. Indeed, it should not be confused with sex, which is more related to what has been made in the present article (i.e., it was considered as a biological factor).

Response: We appreciate this feedback and have modified the column name.

---

## [Editor Report · Decision Letter 2]

16 Oct 2025

Exploratory Analysis of the Accuracy of Age-Based Maximal Heart Rate Equations Across Cardiorespiratory Fitness Levels

PONE-D-25-30112R2

Dear Dr. Martin,

We’re pleased to inform you that your manuscript has been judged scientifically suitable for publication and will be formally accepted for publication once it meets all outstanding technical requirements.

Kind regards,

Stefano Amatori, Ph.D.

Academic Editor

PLOS ONE
---

## [Editor Report · Acceptance letter]

PONE-D-25-30112R2

PLOS ONE

Dear Dr. Martin,

I'm pleased to inform you that your manuscript has been deemed suitable for publication in PLOS ONE. Congratulations! Your manuscript is now being handed over to our production team.

Kind regards,

on behalf of

Prof. Stefano Amatori

Academic Editor

PLOS ONE